# Genomic and phenotypic insights into the expanding phylogenetic landscape of the *Cryptococcus* genus

Marco A. Coelho[1]ᐤ*, Márcia David-Palma[1]ᐤ, Aleksey V. Kachalkin[2,3], Miroslav Kolařík[4], Benedetta Turchetti[5], José Paulo Sampaio[6], Michael J. Wingfield[7], Matthew C. Fisher[8], Andrey M. Yurkov[9], Joseph Heitman[1]*

1 Department of Molecular Genetics and Microbiology, Duke University Medical Center, Durham, North Carolina, United States of America, 2 M.V. Lomonosov Moscow State University, Moscow, Russia, 3 G.K. Skryabin Institute of Biochemistry and Physiology of Microorganisms, PSCBR RAS, Pushchino, Russia, 4 Laboratory of Fungal Genetics and Metabolism, Institute of Microbiology of the Czech Academy of Sciences, Vídeňská, Prague, Czechia, 5 Department of Agricultural, Food and Environmental Sciences, University of Perugia, Perugia, Italy, 6 UCIBIO, i4HB, Departamento de Ciências da Vida, Faculdade de Ciências e Tecnologia, Universidade Nova de Lisboa, Caparica, Portugal, 7 Department of Biochemistry, Genetics and Microbiology, Forestry and Agricultural Biotechnology Institute (FABI), University of Pretoria, Pretoria, South Africa, 8 Department of Infectious Disease Epidemiology, Imperial College London, London, United Kingdom, 9 Leibniz Institute DSMZ-German Collection of Microorganisms and Cell Cultures, Braunschweig, Germany

ᐤ These authors contributed equally to this work.
* marco.dias.coelho@duke.edu (MAC); heitm001@duke.edu (JH)

## Abstract

The fungal genus *Cryptococcus* includes several life-threatening human pathogens as well as diverse saprobic species whose genome architecture, ecology, and evolutionary history remain less well characterized. Understanding how some lineages evolved into major pathogens remains a central challenge and may be advanced by comparisons with their nonpathogenic counterparts. Integrative approaches have become essential for delimiting species and reconstructing evolutionary relationships, particularly in lineages with cryptic diversity or extensive chromosomal rearrangements. Here, we formally characterize six *Cryptococcus* species representing distinct evolutionary lineages, comprising both newly discovered and previously recognized but unnamed taxa, through a combination of phylogenomic analyses, divergence metrics, chromosomal comparisons, mating assays, and phenotypic profiling. Among pathogenic taxa, we formally name *Cryptococcus hyracis* sp. nov., corresponding to the previously characterized VGV lineage within the *C. gattii* complex. In parallel, we describe five saprobic, nonpathogenic species isolated from fruit, soil, and bark beetle galleries, spanning four phylogenetic clades. We identify a strong ecological association with bark beetles for *Cryptococcus porticicola* sp. nov., the only newly described nonpathogenic species with multiple sequenced strains from diverse sites. In this species, we detect strain-level chromosomal variation and evidence of sexual reproduction, along with population-level signatures of recombination. Across the

**Data availability statement:** All primary data are within the paper and its Supporting Information files. Genomic data was deposited in NCBI with accession numbers provided in S6 Table. Genome assemblies and annotations are also available at Figshare (https://doi.org/10.6084/m9.figshare.30375169). The code developed for data analysis and figure generation are publicly available on Zenodo at https://doi.org/10.5281/zenodo.15686462.

**Funding:** This study was supported by the National Institute of Allergy and Infectious Diseases of the National Institutes of Health under awards R01 AI050113-20 (J.H.), R01 AI039115-27 (J.H.), and R01 AI133654-08 (J.H.). We note that these collaborative studies were concluded prior to June 27th, 2025. J.H. is Co-Director and Fellow, and M.C.F. is a Fellow, of the CIFAR program Fungal Kingdom: Threats & Opportunities. Collections leading to the isolation of *C. cederbergensis* were supported entirely and independently by the Harry Oppenheimer Fellowship Award of The Oppenheimer Memorial Trust (to M.J.W.). The funders had no role in study design, data collection and analysis, decision to publish, or preparation of the manuscript.

**Competing interests:** I have read the journal's policy and the authors of this manuscript have the following competing interests: J.H. serves on the Editorial Board of PLOS Genetics. All other authors have declared that no competing interests exist.

genus, chromosome-level comparisons reveal extensive structural variation, including species- and strain-specific rearrangements that may restrict gene flow. We also identify multiple instances of chromosome number reduction, often accompanied by genomic signatures consistent with centromere inactivation or loss of centromeric identity. Comparative metabolic profiling with Biolog phenotype microarrays reveals clade-level differentiation and distinct substrate preferences, which may reflect metabolic divergence and habitat-specific diversification. Notably, we confirm that thermotolerance is restricted to clinically relevant taxa. These findings refine the species-level taxonomy of *Cryptococcus*, broaden its known genomic and ecological diversity, and strengthen the framework for investigating speciation, adaptation, and the emergence of pathogenicity within the genus.

## Author summary

*Cryptococcus* is a genus of fungi that includes both pathogenic species capable of causing life-threatening infections in humans and many environmental species that inhabit soil, fruit, decaying wood, and insect-associated environments with no known link to disease. Here, we formally describe six previously unnamed species based on genome-wide comparisons, genetic divergence metrics, chromosome structure, reproductive behavior, and metabolic profiling. One of these species belongs to the pathogenic lineage, while the others are distinct environmental taxa, including several recovered from bark beetles or their galleries, indicating repeated occurrence in this ecological niche. For one of these insect-associated species, represented by multiple isolates, we detected signatures of sexual reproduction and genetic exchange in natural populations. This study expands the framework for investigating how certain *Cryptococcus* lineages adapt to new environments or evolve traits relevant to pathogenicity. It also emphasizes that continuous environmental sampling remains essential for capturing fungal diversity and illustrates the power of combining genomic and phenotypic data to define species boundaries and uncover lineages that may otherwise remain undetected.

## Introduction

The basidiomycete genus *Cryptococcus* was formally established by Vuillemin in 1901 to describe encapsulated, non-fermentative yeasts that lacked ascospore formation—a key distinction from the genus *Saccharomyces* [1,2]. This taxonomic revision unified two independent discoveries made in 1894: German physicians Otto Busse and Abraham Buschke had isolated a *Saccharomyces*-like yeast from a tibial lesion in a young woman with a chronic granulomatous infection [3], while Italian microbiologist Francesco Sanfelice recovered a similar organism from peach juice, naming it *Saccharomyces neoformans* based on its distinctive colony morphology [4].

Vuillemin's renaming of the organism as *Cryptococcus neoformans* formalized the genus and established its association with human disease.

Over the following decades, *C. neoformans* became recognized as the causative agent of cryptococcal meningoencephalitis, a life-threatening fungal infection with tropism for the central nervous system, particularly in immunocompromised individuals [5]. Its global burden escalated during the HIV/AIDS pandemic of the 1980s and 1990s, when it became a common AIDS-defining illness [2]. Early classifications relied on phenotypic traits such as capsule morphology and serological markers, which were valuable for identification but lacked resolution to resolve cryptic diversity or deeper evolutionary relationships.

A major advance came in 1975, when K.J. Kwon-Chung successfully induced the sexual state of *C. neoformans* (serotype D, now *C. deneoformans*), leading to the description of the teleomorph *Filobasidiella neoformans* [6]. The observed basidia resembled those of the genus *Filobasidium*, prompting the proposal of a new genus to accommodate the sexual state. However, subsequent revisions to the International Code of Nomenclature of algae, fungi and plants (ICNafp)—which moved toward unifying asexual and sexual names—led to the synonymization of *Filobasidiella* under the older name *Cryptococcus*, which has nomenclatural priority. The discovery of the sexual cycle not only clarified the reproductive biology of this organism but also emphasized the ecological and clinical relevance of spore production, which facilitates environmental dispersal and likely exposure of mammalian hosts [7]. Early molecular studies revealed that the genus *Cryptococcus* is polyphyletic, encompassing distantly related yeasts within the class *Tremellomycetes* [8,9]. Subsequent phylogenetic revisions resolved this taxonomic heterogeneity, transferring taxa unrelated to the pathogenic lineage to other genera such as *Naganishia*, *Papiliotrema*, and *Cutaneotrichosporon* [10,11].

Today, *Cryptococcus* is best known for its emblematic pathogenic species, particularly *C. neoformans* and *C. gattii*, which together cause over 112,000 deaths annually, primarily among individuals with advanced HIV/AIDS, with the highest burden in sub-Saharan Africa [5,12]. Owing to their substantial global impact, *C. neoformans* and *C. gattii* have been designated as critical- and medium-priority pathogens, respectively, on the WHO fungal priority list [13]. These two species belong to a broader pathogenic clade presently comprising seven recognized human pathogens: *C. neoformans*, *C. deneoformans*, and five members of the *C. gattii* complex (*C. gattii*, *C. bacillisporus*, *C. deuterogattii*, *C. tetragattii*, and *C. decagattii*) [14]. These taxa differ in virulence, antifungal susceptibility, geographic distribution, and host susceptibility, differences increasingly illuminated through comparative genomics and population-level studies. For instance, *C. deuterogattii* and *C. gattii* sensu stricto frequently cause disease in immunocompetent individuals, the former species being responsible for a well-documented outbreak on Vancouver Island that later expanded to mainland Canada and the Pacific Northwest of the United States [15,16]. In contrast, *C. neoformans*, *C. deneoformans*, *C. bacillisporus*, *C. tetragattii*, and *C. decagattii* are more often associated with infections in immunocompromised hosts [14,17,18]. A recently described, deeply divergent, lineage within the *C. gattii* complex (VGV) has not yet been linked to human disease but exhibits thermotolerance and other traits suggestive of pathogenic potential [19]. Although these species are genetically distinct, hybrids have been documented, particularly between *C. neoformans* and *C. deneoformans* (referred to as AD hybrids, alluding to their different serotypes), and more rarely between *C. neoformans* and members of the *C. gattii* complex. However, these hybrids are typically diploid or aneuploid and exhibit reduced fertility and low spore germination rates (<20%), reflecting postzygotic barriers [14,20,21].

As taxonomic resolution improves, formally naming new *Cryptococcus* species has become both necessary and important. While concerns have been raised about the added complexity for clinical practice [22], failure to recognize distinct phylogenetic lineages can impede diagnosis and obscure clinical and ecological patterns. Even within the *C. gattii* complex, species differ in virulence, antifungal susceptibility, geographic range, and host susceptibility (i.e., immunocompetent vs. immunocompromised) [14,17,18], reinforcing the need for accurate species-level distinctions in both public health and evolutionary research.

Beyond the pathogenic species, *Cryptococcus* harbors several nonpathogenic taxa. Some have been recognized for decades—including *C. wingfieldii*, *C. amylolentus*, *C. depauperatus*, and *C. luteus* [11,23–25]—while others, such as *C. floricola* [26], have been more recently discovered. These species are known exclusively from environmental isolates, typically recovered from soil, plants, or insects, and are presumed saprobic, although *C. depauperatus* and *C. luteus* have been proposed as potential mycoparasites [27–29]. These taxa fall into two distinct clades: one comprising *C. amylolentus*, *C. floricola,* and *C. wingfieldii*, and another represented by *C. depauperatus*. The phylogenetic placement of *C. luteus* remains unresolved due to the lack of genome sequence data. Phylogenetically, the closest known relatives to *Cryptococcus* are found in the genus *Kwoniella*, with both genera forming a well-supported clade within the *Tremellales* (*Tremellomycetes*) [11,30].

Building on this foundation, recent work and sequence database mining have revealed additional deeply divergent *Cryptococcus* lineages that remain undescribed [30–32]. High-quality genome assemblies from these lineages have enabled broader comparative analyses of gene content, chromosomal organization, and mating-type locus (*MAT*) structure, offering new insights into the genomic basis of ecological adaptation and potential pathogenicity [30,32]. Here, we build upon and expand these genomic efforts by integrating newly sequenced genomes with existing data to formally describe six new *Cryptococcus* species representing deeply diverging lineages across multiple clades. Through genome-scale phylogenies, divergence metrics, and chromosomal structural comparisons, we analyze species boundaries, identify evidence of sexual reproduction within a previously uncharacterized bark beetle-associated species, and uncover lineage-specific reductions in chromosome number. We further document phenotypic differentiation across clades, including variation in substrate assimilation profiles, thermotolerance, and melanin production. In doing so, we combine the biological and phylogenetic species concepts to establish a stable, well-supported classification within the genus *Cryptococcus* and provide a broader framework for exploring ecological adaptation and the evolutionary emergence of pathogenicity.

## Results

### Species phylogeny reveals novel *Cryptococcus* diversity and deep evolutionary divergence

We previously analyzed the genomes of 17 *Cryptococcus* species, among which six represent novel lineages that remained undescribed and unnamed [30,32]. These candidate species are distributed across multiple major clades of the genus (defined as clades A to D). To further resolve their phylogenetic placement and support formal species recognition, we reconstructed a robust species phylogeny for *Cryptococcus* based on 2,687 single-copy orthologs (SC-OGs) identified across 39 representative strains (Fig 1A). Where possible, two strains per species were included to assess monophyly and capture intraspecific heterogeneity. For *C. neoformans*, we further incorporated two representatives from each of the major lineages (VNI, VNBI, VNBII, VNII) to reflect known within-species diversity. Strains were chosen based on the availability of complete genome assemblies, inclusion of opposite mating types where possible, and representation of divergent subclades within each lineage as previously defined [33]. Sampling within clade D was also expanded by sequencing, assembling, and annotating the genomes of three additional strains. Two of these (DSM108352 and DSM111204) were sequenced with Oxford Nanopore technology, polished with Illumina reads, and assembled to telomere-to-telomere completeness. A third strain (DSM111205) was sequenced with Illumina only, resulting in a high-quality draft assembly. In addition, the publicly available genome assembly of DSM117830 (GCA_024271845.1) was retrieved from GenBank and annotated in this study.

To infer the species phylogeny, we applied a stringent ortholog selection pipeline (Fig 1C), starting with 3,510 SC-OGs and filtering based on minimum protein length, coverage relative to the *C. neoformans* H99 reference proteome, and parsimony informativeness. A final set of 2,687 high-confidence SC-OGs was retained for phylogenetic inference. Both Maximum Likelihood (ML) and coalescent-based analyses (ASTRAL) produced largely congruent topologies with strong support across most nodes (Fig 1A–B). These analyses robustly recovered the major lineages of *Cryptococcus* pathogens

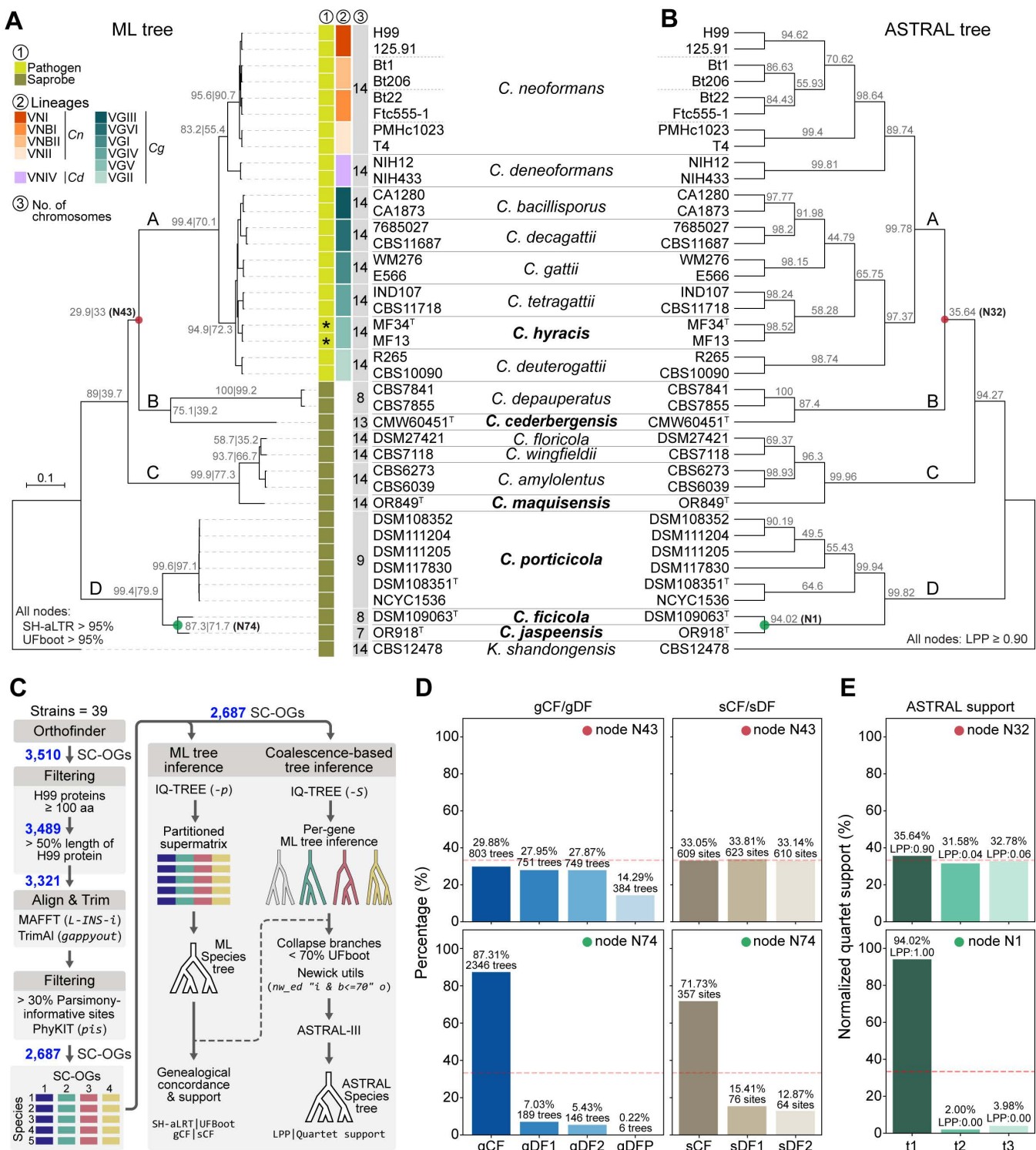

**Fig 1. Phylogenomic relationships and species delimitation within *Cryptococcus*.** (A) Maximum likelihood (ML) phylogeny inferred from a concatenated data matrix comprising protein alignments of 2,687 single-copy orthologs (SC-OGs) across 39 strains, representing 17 *Cryptococcus* species (including six newly proposed species indicated in boldface) and rooted with the outgroup *Kwoniella shandongensis*. Gene and site concordance factors (gCF|sCF) are shown in grey at nodes separating species and major clades. A red circle highlights a conflict-prone node (N43), while a green circle

marks an example of a well-supported node (N74). Support values from SH-aLRT and ultrafast bootstrap (UFboot) exceed 95% for all nodes and were omitted for clarity but are available in the corresponding tree file (https://doi.org/10.5281/zenodo.15686462). Branch lengths are shown as the number of substitutions per site (scale bar). Major clades are labeled as partitions A–D. For reference, lineages designations are provided for *C. neoformans* (*Cn*: VNI, VNBI, VNBII, VNII), *C. deneoformans* (*Cd*: VNIV) and *C. gattii* complex (*Cg*: VGI–VGVI), and chromosome numbers are indicated for each species. Asterisks next to *C. hyracis* strains denote that this species has not yet been associated with human infections. (B) Species tree inferred with ASTRAL, which summarizes gene tree topologies under a multispecies coalescent model, using the same set of 2,687 single-copy orthologs (SC-OGs). Colored circles indicate the corresponding nodes highlighted in panel A, with red marking a conflict-prone node (N32) and green marking an example of a well-supported node (N1). Support values in grey represent the quartet support for the main topology (q1). Local posterior probabilities (LPP) are ≥ 0.90 for all nodes and are not shown for simplicity but are available in the corresponding tree file (https://doi.org/10.5281/zenodo.15686462). Major clades are labeled as partitions A–D. (C) Overview of the phylogenomic workflow. A total of 3,510 single-copy orthologs (SC-OGs) were initially identified using OrthoFinder across 39 genomes. These were filtered based on alignment length, sequence completeness, and phylogenetic informativeness, resulting in 2,687 high-confidence SC-OGs. Protein alignments were used to infer individual gene trees and a concatenated supermatrix. The resulting datasets were analyzed with maximum likelihood (ML) methods (panel A) and the coalescent-based ASTRAL approach (panel B). (D) Gene and site concordance at two representative nodes: N43 (conflict-prone) and N74 (well-supported). Bar plots on the left show gene concordance factors (gCF) and gene discordance factors (gDF1, gDF2, and gDFP), representing the proportion of gene trees that support the main topology (gCF, dark blue), each of the two alternative topologies (gDF1 and gDF2, light blue), or that display conflicting relationships due to paraphyly of one or more clades (gDFP, pale blue). The corresponding site concordance (sCF) and site discordance (sDF1 and sDF2) values are shown on the right, based on site-wise likelihood support for each topology. The red dashed line at 33.3% indicates the expected support under a hard polytomy, where all three resolutions are equally likely due to lack of phylogenetic signal. (E) ASTRAL quartet support for the same nodes shown in panel D: N32 (conflict-prone) and N1 (well-supported). Bar plots show the normalized frequencies of the three possible quartet topologies around each node, based on gene tree topologies. The main topology is shown in dark green, with alternatives in lighter shades. Local posterior probabilities (LPP) for each topology are indicated above the bars. The red dashed line (33.3%) reflects the expected distribution under a hard polytomy.

(clade A)—*C. neoformans*, *C. deneoformans*, and the various *C. gattii* species (sensu lato)—and clearly separated nine saprobic species into three distinct, deeply diverging clades. Several of the newly proposed taxa formed long branches or well-supported monophyletic groups distinct from previously known species, supporting their recognition as separate evolutionary lineages. Clade B includes *C. depauperatus* and strain CMW60451, representing a new species we designate *Cryptococcus cederbergensis* sp. nov., isolated from a bark beetle (*Lanurgus* sp.) in the Cederberg Mountains of South Africa. Clade C comprises *C. floricola*, *C. wingfieldii*, and *C. amylolentus*, along with a divergent lineage represented by a single strain, now named *Cryptococcus maquisensis* sp. nov., previously isolated from soil in Parque Natural da Arrábida, Portugal [34]. Clade D, sister to all other *Cryptococcus* lineages, contains three well-resolved branches, each representing a new species. One of these includes six closely related strains that form a strongly supported monophyletic lineage, designated *Cryptococcus porticicola* sp. nov., named for its occurrence in bark beetle galleries, from which several strains were isolated in Russia and Sweden. An additional strain was isolated from the gut of a bark beetle larva collected in the Czech Republic. The other two branches represent *Cryptococcus ficicola* sp. nov., isolated from a fig in Umbria, Italy, and *Cryptococcus jaspeensis* sp. nov., isolated from soil at Pedreira do Jaspe in Parque Natural da Arrábida, Portugal, with the species name referencing its topographic origin. Additional strain details are provided in S1 Table and in their respective descriptions in the Taxonomy section.

Among the four major clades, the placement of clades B and C relative to clade A was consistent between phylogenetic methods, but this relationship (node N43 in Fig 1A; node N32 in Fig 1B) exhibited comparatively lower gene and site concordance factors (gCF = 29.9%, sCF = 33.05%; Fig 1D), and lower quartet support (35.64%; Fig 1E), indicating persistent conflict at this deep node despite topological agreement. While not a strict cutoff, a threshold near 33% is often applied in both frameworks to assess phylogenetic signal. This value reflects the theoretical expectation that, among the three possible unrooted topologies for a given internode, support below this level suggests that alternative relationships are more frequently favored than the main topology.

To assess whether similar patterns of conflict extended to shallower nodes within clade A, we examined a pruned view of the ML and ASTRAL trees, restricted to species in the *Cryptococcus* pathogenic clade (S1A–S1B Fig). The overall topology was identical between methods, but several internal nodes (N47, N52, N61 in the ML tree) showed relatively low concordance. At these nodes, the main topology was most frequently supported by gene trees (gCF = 26.6–32.7%),

yet a substantial proportion of trees were discordant or paraphyletic (gDFP~46%) (S1C Fig). Likewise, site concordance factors (sCF) only modestly favored the main topology. ASTRAL quartet support for the corresponding nodes (N28, N23, N16; S1D Fig) consistently identified the primary topology (t1) as best supported, with normalized support ranging from 44.8% to 58.3%, and all nodes having local posterior probabilities (LPP) above 0.9. These results suggest that while gene tree conflict is common at these shallow nodes the dominant signal still supports a stable species tree. Such patterns are frequently associated with incomplete lineage sorting (ILS) and recent speciation events [35,36].

## Genome-wide divergence metrics support species boundaries in the *Cryptococcus gattii* complex and the formal recognition of *Cryptococcus hyracis* sp. nov

In addition to the newly identified nonpathogenic species, we formally recognize a previously described but unnamed lineage within the *C. gattii* complex. This lineage, referred to as VGV by Farrer et al. [19], was discovered in the Central Zambezian Miombo Woodlands of Zambia and isolated from tree holes and hyrax middens (stratified fecal and urinary deposits formed at communal latrines). In that study, VGV was shown to be genetically distinct from other *C. gattii* lineages (VGI–VGIV), based on both population genetic analyses and its placement as a monophyletic group in genome-wide phylogenies. Species names for all major molecular types (VGI–VGIV and VGVI) had been formally proposed by Hagen et al. in 2015 [14], establishing a precedent for detailed taxonomic recognition within the complex. Because a subsequent publication expressed caution regarding the stability and interpretation of species boundaries in this group [22], the VGV lineage remained unnamed despite clear evidence of genetic divergence. To maintain consistency with the framework proposed by Hagen et al. [14,17], and reflect our view that recognizing distinct species within the *C. gattii* complex is advantageous for genomic and ecological clarity, we propose to designate the VGV lineage as *Cryptococcus hyracis* sp. nov., in reference to its frequent association with hyrax-inhabited environments. In our phylogenetic analyses, *C. hyracis* formed a well-supported clade distinct from its closest relative, *C. tetragattii* (VGIV) (Figs 1A–1B and S1A–S1B). The two species were resolved as sister groups in the ASTRAL tree with moderate quartet support (58.3%, node N23; S1B and S1D Fig), and gene concordance fell just below the 33% threshold for inferring a dominant phylogenetic signal (gCF = 32.7%, node N52; S1A and S1C Fig), again pointing to potential ILS or rapid divergence.

To complement the phylogenetic analyses and further evaluate species boundaries, we calculated genome-wide divergence metrics including average nucleotide identity (ANI), average amino acid identity (AAI), and digital DNA–DNA hybridization (dDDH) across species. These metrics, widely adopted in prokaryotic taxonomy [37], are increasingly applied to yeast systematics under the taxogenomic framework to quantify genetic relatedness beyond traditional barcode sequences [38]. ANI measures nucleotide-level similarity between homologous genomic regions, with values above ~95–96% typically indicating conspecificity in prokaryotes. Similarly, dDDH, originally developed to approximate wet-lab DNA–DNA hybridization, estimates intergenomic relatedness using BLAST-based comparisons and retains the conventional 70% threshold for species delineation [39]. However, the thresholds developed for prokaryotes are not universally applicable to fungi without prior benchmarking, and their application in yeast remains underdeveloped. Broader application of these and other genomic indices will likely require integrating additional criteria, such as sexual compatibility and reproductive isolation under the biological species concept. While limited in scope and largely focused on ascomycetous yeasts, initial studies suggest these metrics can be informative when aligned with phylogenetic and phenotypic data [38,40].

To contextualize these metrics for species delimitation and ensure their appropriate application to basidiomycete fungi, *C. neoformans* was selected as a benchmark. This well-circumscribed species exhibits a documented sexual cycle and reproductive isolation. Genome comparisons included eight *C. neoformans* strains, with two representatives from each of the major lineages (VNI, VNII, VNBI, and VNBII), providing a reference for intraspecific variation. Although these lineages exhibit substantial population structure and limited recombination between them, they are recognized as a single species

[33]. Thus, the lowest similarity observed among these strains provides a conservative upper bound for intraspecific genomic divergence within *Cryptococcus*. The minimum ANI and AAI values observed among *C. neoformans* lineages were 97.7% and 97.4%, respectively, corresponding to divergences of 2.3% and 2.6% (Fig 2A–2C and S2 Table). Similarly, to provide a reference point for the lower bound of interspecific divergence, we used the species pair *C. bacillisporus* and *C. decagattii,* which are considered taxonomically distinct [14,18] despite relatively high genomic similarity. The highest observed ANI and AAI values between strains of these two species were 96.5% and 95.5%, respectively, corresponding to divergences of 3.5% and 4.5%. These values fall below the lowest ANI and AAI observed in any intraspecific comparison in our dataset (Fig 2C and S2 Table), reinforcing the interpretation of *C. bacillisporus* and *C. decagattii* as a species-level boundary case.

Comparatively, pairwise analysis between *C. hyracis* and its sister species *C. tetragattii* yielded ANI values ranging from 95.32% to 95.51% and AAI values from 95.21% to 95.61%, depending on the strain pair. These are lower than the values observed within *C. neoformans* lineages (97.7% ANI and 97.4% AAI) and also lower than those between *C. bacillisporus* and *C. decagattii* (96.3–96.5% ANI; 95.4–95.5% AAI), the most closely related interspecies pair in our dataset (Fig 2A–2B and S2 Table). The ANI vs. AAI scatterplot (Fig 2C) reinforced this pattern: *C. hyracis*–*C. tetragattii* comparisons fell well below the intraspecies cluster and were among the most similar interspecies comparisons, yet still more divergent than *C. bacillisporus*–*C. decagattii*. The strong correlation between ANI and AAI across all comparisons (Spearman $\rho = 0.8966$, $P < 3.9e{-}237$; Pearson $r = 0.9912$, $P < 1e{-}300$) supports the consistency of these metrics in capturing species-level genomic divergence.

Estimates based on dDDH further supported the distinctiveness of *C. hyracis*. Pairwise dDDH values between *C. hyracis* and *C. tetragattii* ranged from 62.1% to 62.8%, with corresponding probabilities of exceeding the 70% species threshold between 59.1% and 61.3% (Fig 2D–2F and S2 Table). For comparison, *C. bacillisporus* and *C. decagattii* yielded a dDDH value of approximately 69.3%, just below the conventional cutoff, but with a greater than 75% probability that the true value exceeds the 70% threshold. Nonetheless, these interspecific values remain significantly lower than those observed between *C. neoformans* strains, where dDDH ranges from 79% (probability ≥70% = 89.75%) to 99.1% (probability ≥70% = 98.07) (Fig 2E and S2 Table). This suggests that *C. bacillisporus* and *C. decagattii* lie near the boundary of species-level divergence. In contrast, *C. hyracis* and *C. tetragattii* show a more pronounced level of genomic separation. Together, these results support the recognition of *C. hyracis* as a distinct species, building on previous work by Farrer et al. [19] that first identified it as a divergent lineage within the *C. gattii* complex.

Although all currently available strains of *C. hyracis* are of the *MAT*α mating type and no sexual cycle has been observed thus far, we tested whether *C. hyracis* could engage in mating with strains from other *C. gattii* (sensu lato) species. Our assays showed that *C. hyracis* is capable of forming sexual structures in pairings with *MAT***a** strains of *C. bacillisporus*, *C. decagattii*, *C. tetragattii,* and *C. deuterogattii* (S2 Fig). Notably, strain MF13 consistently exhibited stronger mating responses than MF34, indicating possible strain-level differences in mating competence, consistent with prior observations of intraspecific variation and their placement in distinct subclades [19]. While the fertility and ploidy status of resulting progeny were not assessed here, these preliminary results suggest that *C. hyracis* retains the ability to mate with other species in the complex. A more systematic investigation of post-zygotic outcomes will be needed to assess reproductive compatibility and isolation across the *C. gattii* complex and to test species boundaries under the biological species concept.

## Chromosomal rearrangements and karyotypic diversification in clade A species

Structural variation at the chromosomal level provides a complementary axis of genome divergence and may offer important insights into reproductive isolation and the emergence of species boundaries. All pathogenic *Cryptococcus* species (clade A) maintain a karyotype of 14 chromosomes (Fig 1A) [32], and previous studies based on earlier genome assemblies have identified chromosomal rearrangements between *C. neoformans*, *C. deneoformans*, and members of the

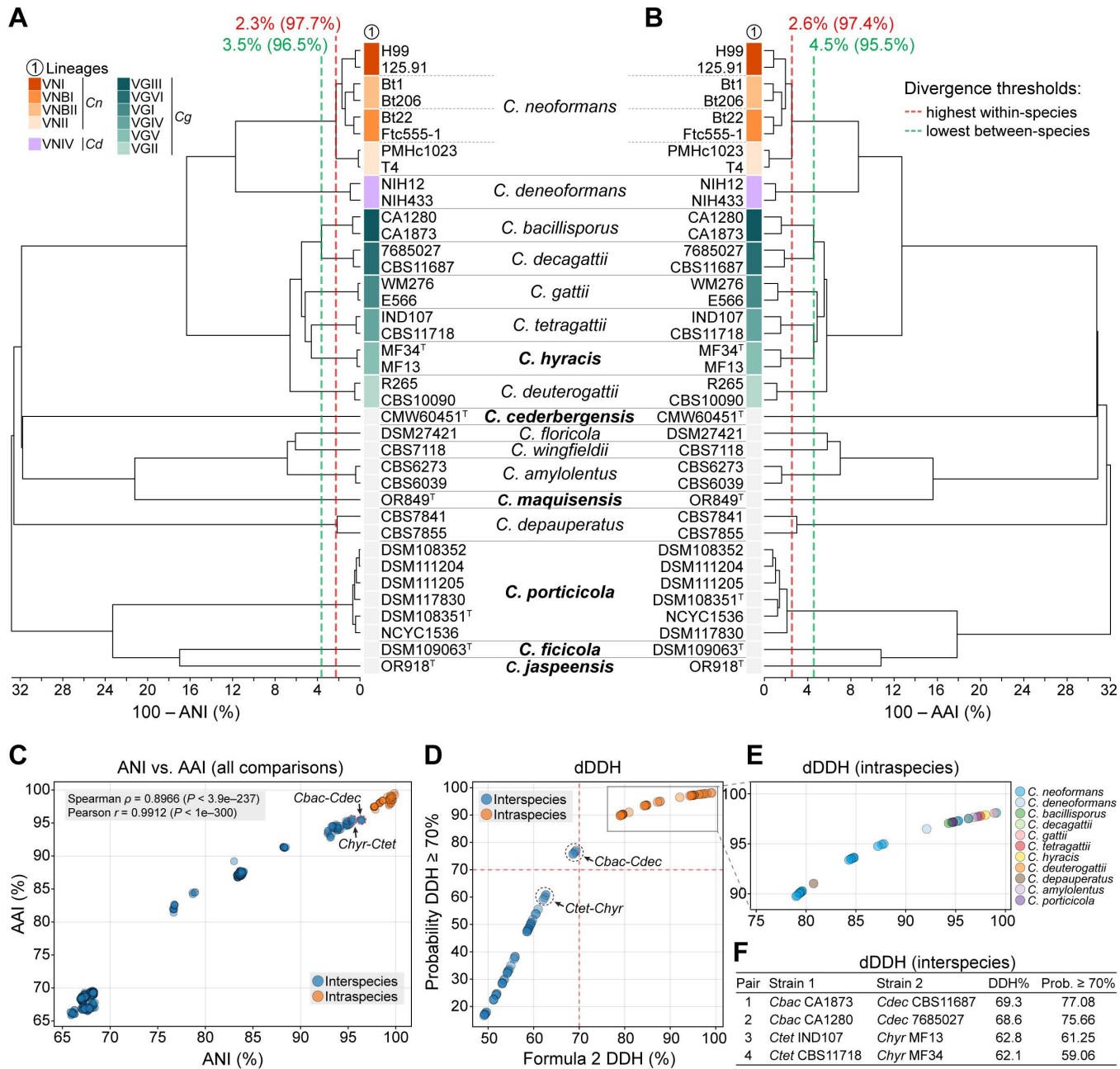

**Fig 2. Genomic divergence and species boundaries among *Cryptococcus* species.** (A–B) UPGMA clustering based on pairwise sequence divergence, calculated as 100 – Average Nucleotide Identity (ANI) in panel A and 100 – Average Amino Acid Identity (AAI) in panel B. Vertical dashed lines indicate divergence thresholds: red marks the highest observed divergence within a species (i.e., the lowest ANI or AAI between *C. neoformans* lineages), and green marks the lowest observed divergence between species (i.e., the highest ANI or AAI between *C. bacillisporus* and *C. decagattii*). (C) Scatterplot comparing ANI and AAI values across all pairwise genome comparisons. Intraspecies comparisons are shown in orange and interspecies comparisons in blue. The two interspecies pairs with the highest genomic similarity are highlighted: *C. bacillisporus* (*Cbac*) vs. *C. decagattii* (*Cdec*), and *C. hyracis* (*Chyr*) vs. *C. tetragattii* (*Ctet*). Pearson and Spearman correlation coefficients indicate a strong positive relationship between ANI and AAI values across all comparisons. (D) Digital DNA–DNA hybridization (dDDH) estimates for pairwise genome comparisons. The x-axis shows values derived from *Formula 2* of the Genome-to-Genome Distance Calculator (GGDC), which estimates the expected wet-lab DDH percentage based on genome BLAST comparisons. The y-axis shows the probability that the true DDH value exceeds 70%, a widely accepted threshold for species demarcation. Dashed lines indicate the 70% threshold for both axes. Intraspecies comparisons are shown in orange and interspecies comparisons in blue. (E) Zoomed-in view on intraspecies dDDH comparisons, grouped by species. (F) Table of dDDH values for select interspecies comparisons that approach

or exceed the 70% threshold, including the genome pairs with the highest interspecific scores. *C. bacillisporus–C. decagattii* comparisons fall just below the 70% cutoff but have high probabilities (>75%) of exceeding it, indicating borderline genomic divergence despite their current recognition as separate species. In contrast, *C. tetragattii–C. hyracis* comparisons fall well below both thresholds, supporting their recognition as distinct species.

*C. gattii* complex [19,41–43]. However, these analyses were limited to a smaller set of species or relied on lower-resolution assemblies, leaving the extent and pattern of large-scale chromosomal rearrangements across these species not fully characterized. To address this, we first performed comparative synteny analyses across clade A species using *C. neoformans* strain 125.91 as a reference. This strain was selected over the more commonly studied H99, which carries a private translocation [44] and therefore does not represent the typical karyotype of the species (S3A–S3B Fig). Interspecies differences are summarized in Fig 3 with one representative strain per species. S3 Fig presents additional karyotype comparisons among *C. neoformans* and *C. deneoformans* strains, while S4 Fig includes all analyzed strains from the *C. gattii* complex to capture both species- and strain-specific variation.

Our comparisons revealed multiple species- and strain-specific structural differences, including large translocations and inversions. Three translocations shared by *C. bacillisporus* and *C. decagattii* appear to involve centromeric regions of chr. pairs 1–2, 4–5, and 12–13 in *C. neoformans* 125.91 (Fig 3A–3B), consistent with intercentromeric recombination (ICR) as a known mechanism of chromosome reshuffling in *Cryptococcus* [43,45]. Two of these events (1–2 and 4–5) are older, as they are present in all analyzed *C. gattii* (sensu lato) species, suggesting that they originated in their last common ancestor. In contrast, the third event (chr. pair 12–13) is absent in *C. gattii* (sensu stricto), *C. tetragattii*, *C. hyracis*, and *C. deuterogattii*, indicating that it occurred later, in the lineage leading to *C. bacillisporus* and *C. decagattii*.

Additional translocations were identified among *C. gattii* lineages (S4 Fig). For example, chrs. 1 and 11 in *C. deuterogattii* appear to derive from a translocation between corresponding chrs. 3 and 9 of *C. gattii* WM276, which are conserved in related species and likely represent the ancestral state (anc3 and anc9) (Figs 3A and S4B, event a). Two further rearrangements involving anc12–anc13 and anc8–anc10 define the common ancestor of *C. bacillisporus* and *C. decagattii* (S4B Fig, events b and c). In *C. decagattii*, one of the resulting chromosomes (corresponding to extant *C. bacillisporus* chr. 5) underwent a subsequent translocation with anc7, giving rise to extant chrs. 4 and 9 in *C. decagattii* (S4B Fig, event d). We also detected two strain-specific translocations: one in *C. decagattii* CBS11687 (between anc1 and anc11; S4B Fig, event e) and another in *C. tetragattii* CBS11718 (between chrs. anc6 and anc13; S4B Fig, event f). As a result, *C. bacillisporus* differs from *C. decagattii* by one fixed translocation (relative to strain 7685027) or two (relative to strain CBS11687), which further impose postzygotic reproductive barriers.

In contrast, *C. hyracis* strain MF34 and *C. tetragattii* IND107 share a largely collinear karyotype, differing only by a single inversion on chr. 2. Similarly, *C. neoformans* 125.91 and *C. deneoformans* NIH433 differ only by two large inversions (on chrs. 1 and 9; Fig 3), with otherwise conserved chromosomal structures. Although these species pairs show limited karyotypic divergence, their clear sequence-level divergence, particularly in the case of *C. neoformans* and *C. deneoformans*, supports their recognition as distinct species, suggesting that large-scale structural changes may not always accompany speciation.

Together, these results reveal that karyotypic diversification across clade A involves a mixture of lineage-specific rearrangements and ongoing structural changes at the strain level. While fixed chromosomal rearrangements likely reinforce reproductive isolation in some lineages (such as between *C. bacillisporus* and *C. decagattii*) their absence in others, such as *C. hyracis* and *C. tetragattii*, underscores a complex and potentially asynchronous relationship between genome structural variation and species divergence in *Cryptococcus*.

## Intraspecific variation and evidence of recombination in *Cryptococcus porticicola*

Among members of newly described clade D, *Cryptococcus porticicola* is the only species for which multiple strains are currently available and have been sequenced, enabling analyses of intraspecific genome variation, chromosomal

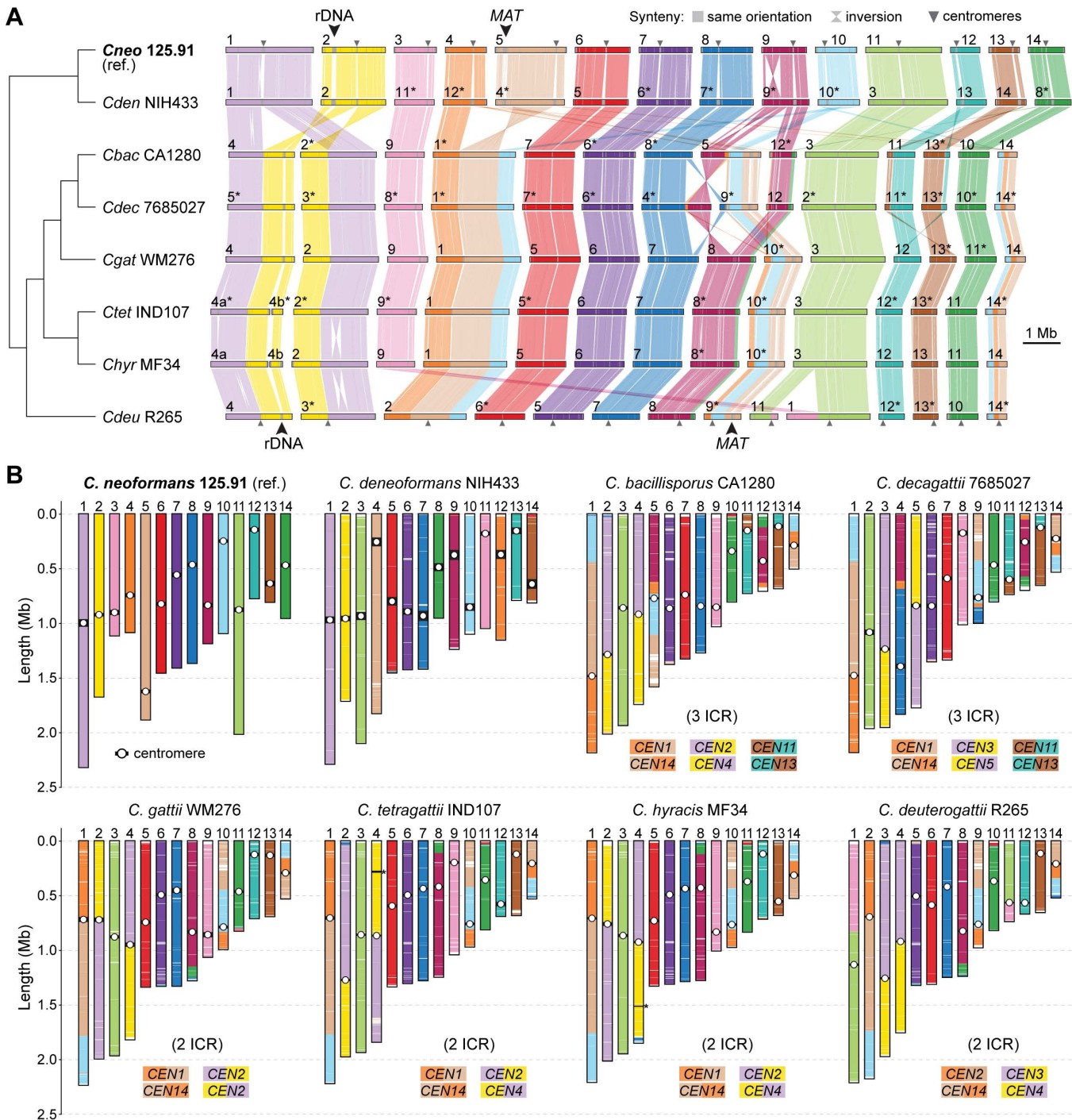

**Fig 3. Chromosomal rearrangements among clade A pathogenic *Cryptococcus* species.** (A) Synteny comparisons between *C. neoformans* strain 125.91 (reference) and representative strains from 7 other clade A species (8 species total). Rearrangements, including large inversions and transloca-tions, were identified using the ntSynt toolkit and visualized with ntSynt-viz. All species maintain 14 chromosomes, but some differences in chromosomal structure are evident. Chromosomes were reordered to maximize collinearity, and those inverted relative to their original assembly orientations are marked with asterisks. The positions of the mating-type locus (*MAT*) and the rRNA locus are also indicated. (B) Schematic comparison of chromosome organization across the same strains, highlighting large-scale translocations and centromere locations. Chromosomes are color-coded according to their correspondence with the *C. neoformans* reference. Putative intercentromeric recombination (ICR) events are indicated where synteny breakpoints coin-cide with centromeres. Asterisks denote chromosomes comprising two contigs interrupted at the rDNA array. Diagrams were generated using a custom Python script (available at https://doi.org/10.5281/zenodo.15686462) based on pairwise alignments from minimap2.

structural dynamics, and reproductive potential. To explore these aspects, we first conducted pairwise genome comparisons among the six sequenced strains revealing high overall similarity, with ANI values ranging from 99.29% to 99.94% and dDDH values from 94.6% to 97.2% (Fig 2E and S2 Table), confirming that all isolates belong to the same species despite some geographic separation.

Next, we conducted mating assays between strain pairs predicted to be compatible based on sequence differences at both the pheromone/receptor (*STE3*) and homeodomain (*HD*) mating-type loci (Fig 4A). These two unlinked *MAT* loci were previously characterized and shown to follow a tetrapolar configuration in *C. porticicola* and other clade D species [30]. Among the tested combinations (S5 Fig), only the cross between strains DSM108352 (*a1b3*) and NCYC1536 (*a2b2*) sporadically produced visible hyphae and putative sexual structures embedded in the agar, resembling basidia (Fig 4B–4C). These structures emerged only after extended incubation (~1 month) and were not consistently observed across replicates. Although their identity could not be definitively confirmed and may represent chlamydospore-like cells, their morphology is more consistent with sexual development observed in related *Kwoniella* species [46–48]. These structures were observed on V8 medium pH 5.0, whereas growth on MS medium led to hyphal formation but without basidial-like clusters (S5D Fig). Notably, some hyphal segments appeared to contain unfused clamp connections (Fig 4C), potentially indicating a monokaryotic mycelium.

To further investigate genetic diversity and potential recombination in *C. porticicola*, we analyzed genome-wide single nucleotide polymorphisms (SNPs) across the six available strains. High-confidence SNPs were identified with Snippy pipeline using DSM108351 as the reference, generating a core genome SNP matrix. A neighbor-net network constructed from this alignment revealed reticulate relationships, suggesting conflicting phylogenetic signals and potential recombination events (Fig 4D). This interpretation is further supported by the Pairwise Homoplasy Index (PHI) test implemented in SplitsTree, which returned a highly significant result ($P < 1e-6$), suggesting that the observed pattern of site incompatibilities is unlikely to arise under a strictly clonal (non-recombining) model. Principal component analysis (PCA) also indicated genetic differentiation among strains, with DSM108352 and NCYC1536—the only cross producing putative sexual structures—showing the greatest separation along PC1, which accounts for the largest proportion of genetic variance (Fig 4E).

To assess patterns of historical recombination more directly, we examined linkage disequilibrium (LD). LD decay analysis, conducted after applying a minor allele frequency (MAF) threshold of 0.167 to reduce noise from rare variants, revealed a clear decline in $r^2$ values with increasing physical distance between SNPs, consistent with recombination gradually breaking down associations between alleles (Fig 4F). Based on the smoothed decay curve, $r^2$ dropped to 50% of its initial maximum (LD50) at approximately 13.9 kb. Taken together, and with the caveat of limited sample size, these findings indicate that *C. porticicola* undergoes some degree of recombination, consistent with the morphological features observed in mating assays that resemble, but do not definitively confirm, sexual development.

Finally, to assess whether chromosomal structural variation parallels the sequence-level diversity observed in *C. porticicola*, we compared chromosome-scale assemblies from four strains spanning the phylogenetic breadth of the species (Fig 4A and 4G). All strains possess 9 chromosomes, indicating a conserved karyotype across the species. Synteny comparisons, using DSM108351 as the reference, revealed a largely conserved chromosomal architecture between DSM108351 and NCYC1536, which are more closely related. Nevertheless, several intraspecific rearrangements were identified (Fig 4G). Two reciprocal translocations are shared between DSM108352 and DSM111204, suggesting these events arose in their common ancestor. Additionally, DSM111204 harbors a third, strain-specific rearrangement with centromere-proximal breakpoints, consistent with an intercentromeric recombination (ICR) event (Fig 4H). These results demonstrate that *C. porticicola* harbors detectable chromosomal structural variation among strains, including translocations that may have functional consequences during meiosis. Notably, the rearrangements observed between DSM108352 and NCYC1536 (the only pair producing putative sexual structures) could contribute to postzygotic isolation through segregation defects, potentially explaining the rarity and incompleteness of sexual development. These findings suggest that *C. porticicola* may comprise distinct populations (e.g. geographic or host-related) with established genomic and structural divergence.

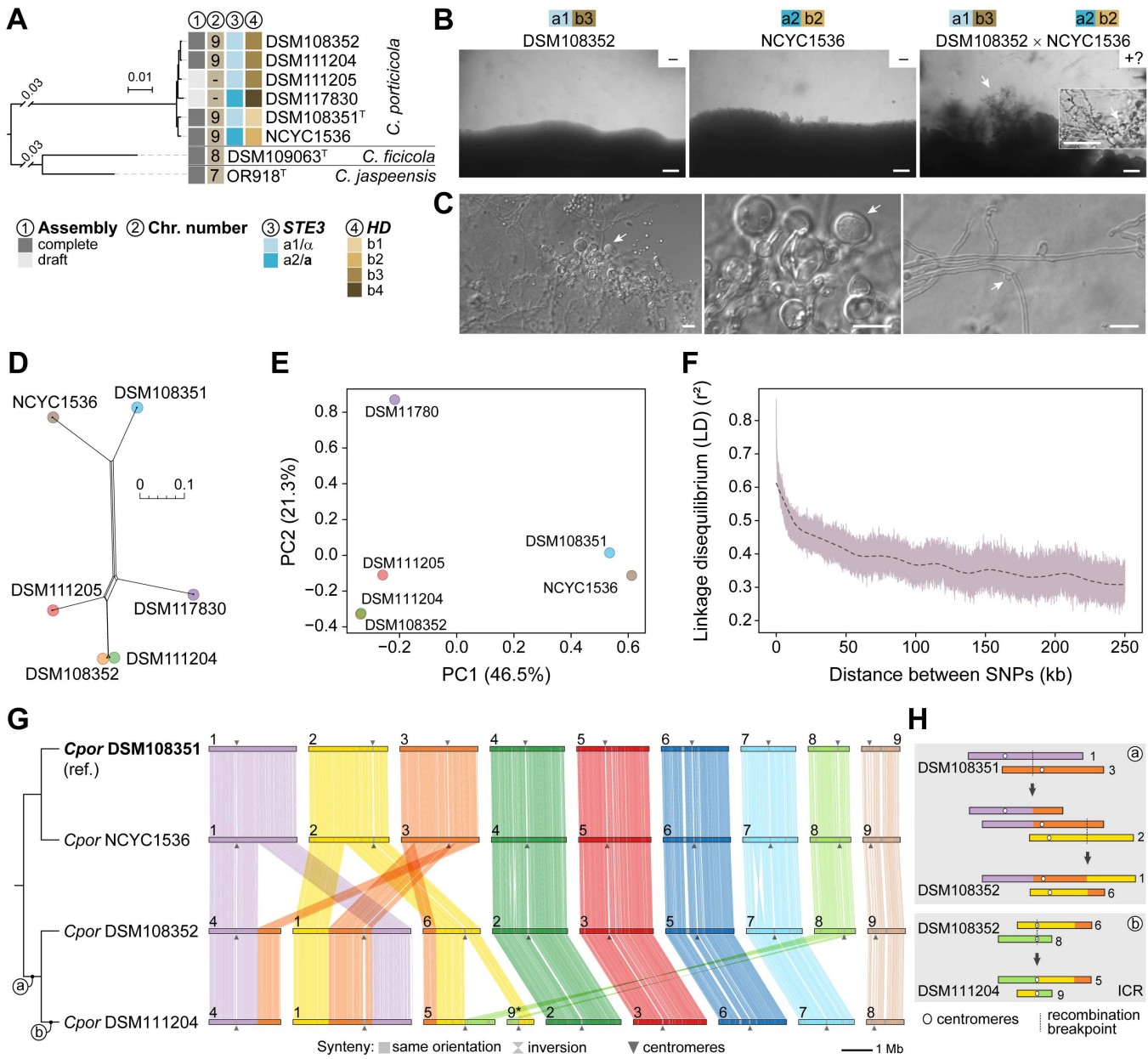

**Fig 4. Mating, recombination signatures, and chromosomal structural variation in *Cryptococcus porticicola* (A) Maximum likelihood phylogeny of clade D species (pruned from Fig 1), highlighting *C. porticicola*, *C. ficicola*, and *C. jaspeensis*.** Labels indicate genome assembly status (complete or draft), chromosome number inferred from telomere-to-telomere assemblies, mating pheromone receptor allele (*STE3* a1/α or *STE3* a2/**a**), and *HD* allele identity (b1–b4) inferred from sequence comparisons. (B) Mating assay between *C. porticicola* strains DSM108352 (*a1b3*) and NCYC1536 (*a2b2*), showing hyphal formation and putative sexual structures. Solo cultures are shown on the left and center panels; the cross is shown on the right, where hyphal formation is observed embedded in the agar. Basidia-like structures are visible and resemble sexual development in *Kwoniella* species. Assays were performed on V8 medium (pH 5.0) at room temperature in the dark for 1 month. Scale bars: 200 µm (50 µm in the inset). (C) Light microscopy images of hyphal and putative sexual structures recovered from the mating plate shown in panel B. The left panel shows hyphae and rounded cells resembling basidia (arrows), with a zoomed-in view in the middle panel. The right panel highlights a hyphal segment with an apparently unfused clamp connection (arrow). Scale bars: 10 µm. (D) Neighbor-Net phylogenetic network constructed from core SNPs showing signals of conflicting phylogenetic relationships (reticulation), consistent with recombination. (E) Principal component analysis (PCA) of core SNP data showing genetic structure among the six strains. The first two principal components explain 46.5% and 21.3% of the total variance, respectively. (F) Linkage disequilibrium (LD) decay curve showing mean pairwise *r²* values (minor allele frequency, MAF ≥ 0.167) as a function of physical distance between SNPs. Dashed line shows lowess-smoothed trend. (G) Chromosomal synteny comparisons across *C. porticicola* employing strain

DSM108351 as reference. Each block represents a chromosome; ribbons denote orthologous regions as determined by the ntSynt toolkit. Chromosomes were reordered to maximize collinearity, and those inverted relative to their original assembly orientations are marked with asterisks. (H) Detailed view of intraspecific chromosomal rearrangements in *C. porticicola*. Centromeres (circles) and synteny breakpoints are shown for each rearrangement. ICR: intercentromeric recombination.

## Chromosome number reduction and centromere dynamics in clade D species

The chromosomal variation identified within *C. porticicola* prompted us to investigate broader trends in karyotype evolution across clade D. All three newly described species in this clade exhibit marked reductions in chromosome number relative to *Cryptococcus* species of clades A and C, which maintain the ancestral 14-chromosome karyotype inferred for the genus [32]. Specifically, *C. porticicola* has 9 chromosomes, while *C. ficicola* and *C. jaspeensis* have only 8 and 7 chromosomes, respectively (Fig 5A).

To elucidate the genomic changes underlying these reductions, we compared chromosome-scale assemblies using *C. porticicola* DSM108351 as a reference and mapped centromere positions across all available strains of clade D. In *Cryptococcus*, centromeres are regional rather than point-like, usually spanning tens of kilobases and characterized by a lack of coding sequences, enrichment in transposable elements—particularly long-terminal repeat (LTR) retrotransposons— and high levels of 5-methylcytosine (5mCG) DNA methylation [41,45,49–51]. DNA methylation calls were generated from Oxford Nanopore sequencing data obtained either in this work or from previously published datasets [30,32]. In *C. jaspeensis*, the region syntenic to *CEN7* of *C. porticicola* and *C. ficicola* lacks hallmark centromeric features and likely no longer functions as a centromere (Figs 5B and S6F). Similarly, both *C. ficicola* and *C. jaspeensis* contain additional presumably inactivated centromeres (*iCENs*) corresponding to *CEN9* and *CEN8* of *C. porticicola*. These *iCENs* retain conserved flanking genes but are depleted of LTR retrotransposons and show only residual 5mCG methylation, consistent with erosion of centromeric identity [52,53] while preserving local gene order (Figs 5D, panel III, and S6G).

Notably, the number of *iCENs* in *C. ficicola* (two) and *C. jaspeensis* (three) exceeds the reduction in chromosome number relative to *C. porticicola*, implying that centromere inactivation alone does not fully explain their current karyotypes. A conserved region on chr. 4 in *C. ficicola* and chr. 5 in *C. jaspeensis* exhibits clear centromeric hallmarks, including TE accumulation and 5mCG enrichment, while the syntenic region in *C. porticicola* displays only spatially restricted 5mCG enrichment and a small number of TEs but does not coincide with any predicted centromere (Fig 5C and 5D, panel II). These findings point to a centromere repositioning event in the ancestor of *C. ficicola* and *C. jaspeensis*, although whether this region represents a newly co-opted centromere or the retention of an ancestral one that became inactive in *C. porticicola* remains uncertain. Direct functional validation will be required to confirm centromeric activity. Importantly, gene order is preserved across these chromosomes, suggesting that centromere repositioning, if it occurred, did so without major local rearrangement. Together, these findings suggest that chromosome number reduction in clade D species, potentially contributing to species diversification, is linked to structural rearrangements followed by centromere inactivation.

## Variation in rRNA gene organization across *Cryptococcus* species

In addition to chromosome number changes, variation in ribosomal RNA (rRNA) gene organization represents another axis of genome structural divergence in *Cryptococcus* [32]. To explore this in more detail, we applied a unified pipeline that integrates rRNA gene prediction, clustering detection, and statistical enrichment tests (see Material and Methods) to survey the chromosomal arrangement of 5S, 5.8S, 18S, and 28S rRNA genes across 20 chromosome-level genome assemblies representing all major clades and species (Figs 6 and S7).

Fully clustered rDNA arrays (defined as tandem repeats of the canonical 28S–5.8S–18S–5S gene units) were observed exclusively in pathogenic species from clade A. In contrast, other species exhibited unclustered architectures, with the 5S gene physically separated from the 28S–5.8S–18S unit and exhibiting lineage-specific variation in

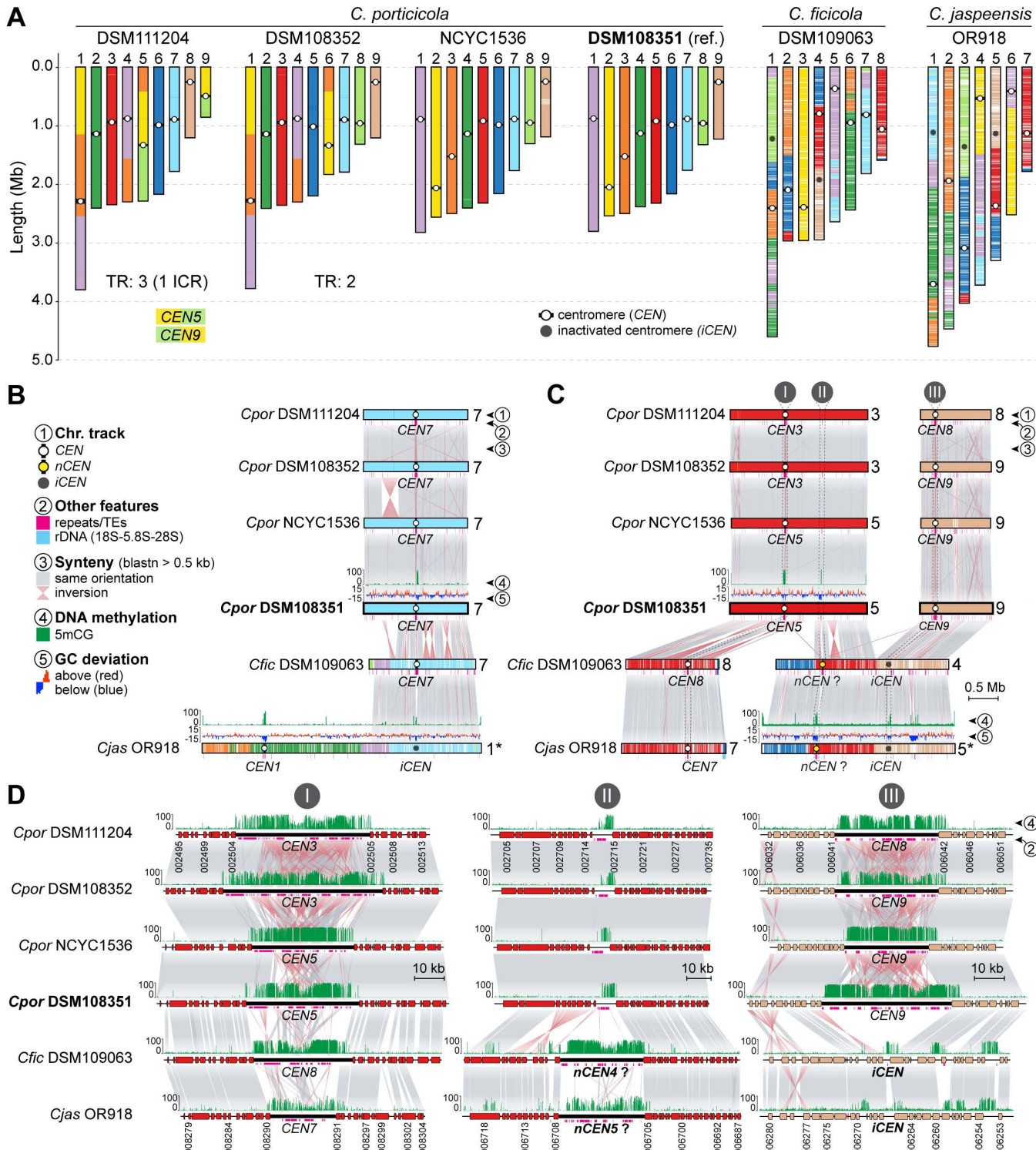

**Fig 5. Karyotype evolution and centromere dynamics in clade D *Cryptococcus* species.** (A) Chromosome-length plots for six telomere-to-telomere assemblies representing *C. porticicola*, *C. ficicola*, and *C. jaspeensis*. Colors indicate synteny blocks, with *C. porticicola* DSM108351 as the reference. Chromosome number differs across species (9, 8, and 7, respectively), and multiple structural changes are observed, including species- and strain-specific translocations (TR). Circles mark predicted centromeres: white for active centromeres (*CEN*) and dark gray for putative inactivated centromeres

(*iCEN*). For *C. porticicola*, the total number of translocations (including those consistent with intercentromeric recombination, ICR) is noted below each strain. (B) Example of centromere loss in *C. jaspeensis*. A region syntenic to *CEN7* in *C. porticicola* and *C. ficicola* appears inactivated in *C. jaspeensis* OR918 (*iCEN* on chr. 1), but still shows 5mCG methylation, (see S6F Fig for a zoomed-in view of this region). Track descriptions are provided on the key on the left. (C–D) Synteny comparisons showing putative centromere inactivation and repositioning in *C. ficicola* and *C. jaspeensis*. In panel C, *CEN5* of *C. porticicola* DSM108351 aligns with *CEN8* in *C. ficicola* and *CEN7* in *C. jaspeensis*, suggesting a shared ancestral origin (zoomed view in panel D, region I). By contrast, *CEN9* in *C. porticicola* appears to have been inactivated (*iCEN*) in both *C. ficicola* and *C. jaspeensis*, coinciding with reduced transposable element content and diminished 5mCG methylation signal, and conservation of flanking genes aside from a minor inversion (panel D, region III). The predicted centromere locations in chr. 4 of *C. ficicola* and chr. 5 of *C. jaspeensis* lack orthology with any of *C. porticicola* centromeres, but the corresponding region in *C. porticicola* exhibits low levels of transposons and 5mCG methylation (panel D, region II). This region may represent a repositioned or newly co-opted centromere (labeled *nCEN*) in the common ancestor of *C. ficicola* and *C. jaspeensis*, or alternatively may have harbored an ancestral centromere that was subsequently lost or inactivated in *C. porticicola*.

chromosomal distribution. Among clade C species, the 5S genes remained significantly enriched on a single chromosome (chi-squared test, $P < 0.0001$ for all species; Figs 6B and S7, and S3 Table), suggesting that despite dispersion, the ancestral chromosomal context of the rDNA array may be partly retained. For instance, in *C. floricola*, over 50% of all 5S genes are located on a single chromosome (standardized residual > +11; S3 Table), which also harbors the canonical rDNA array. By contrast, in other clade C species (*C. wingfieldii* and *C. maquisensis*), the 28S–18S–5.8S array appears relocated to alternate chromosomal ends, consistent with interchromosomal mobility potentially mediated by subtelomeric recombination (S7 Fig).

Clade D species also display unclustered rRNA gene organization but with differing 5S patterns. In *C. porticicola*, 5S genes are distributed relatively evenly across chromosomes (chi-squared test, $P = 0.87$; S3 Table). Conversely, *C. ficicola* and *C. jaspeensis* exhibit significant 5S gene enrichment on specific chromosomes (chi-squared test, $P = 0.0006$ and 0.011, respectively) coupled with a strong subtelomeric bias (Fisher's exact, $P < 1.12e^{-37}$ and $P < 3.03e^{-15}$, respectively; S3 Table). In both species, the majority of 5S genes reside near chromosome ends, suggesting relocation to subtelomeric regions, possibly through recombination or transposition. Together, these results indicate that 5S rRNA gene organization varies substantially across *Cryptococcus*, with clustered arrays retained in pathogenic species and more dispersed patterns in nonpathogenic lineages, suggesting lineage-specific shifts in rDNA architecture alongside broader chromosomal evolution.

## Phenotypic differentiation among *Cryptococcus* clades

To examine whether genomic divergence across the *Cryptococcus* genus is accompanied by phenotypic differentiation, we performed comparative growth and metabolic profiling across 16 of the 17 recognized species. Eighteen strains were tested, including duplicate representatives for *C. porticicola* and *C. amylolentus* (Fig 7 and S4 Table). *C. depauperatus* was excluded due to its markedly slower, filamentous growth, which was incompatible with standardized assay conditions.

We first assessed the metabolic capacity of all strains with Biolog Phenotype MicroArrays, testing growth across 153 substrates (see Methods for details). Of these, 112 yielded variable responses across the 18 strains and were retained for comparative analyses (S4 Table). Hierarchical clustering of the binarized profiles (i.e., positive or negative growth) revealed clade-level groupings, indicating that species within the same clade tend to share more similar metabolic profiles (Fig 7A). However, the branching order in the dendrogram does not mirror the species phylogeny (Fig 1), suggesting that while combination of metabolic traits can discriminate clades, they may reflect ecological adaptation or functional convergence rather than shared evolutionary history. Principal component analysis (PCA) of the same dataset (Fig 7B) supported broad clade separation along PC1 (23.7%) and PC2 (10.9%). The only exception was *C. cederbergensis* (clade B), which clustered within the 95% confidence ellipse of clade C, suggesting metabolic similarity despite phylogenetic distance. This pattern may, however, change with the discovery and characterization of additional clade B species in the future.

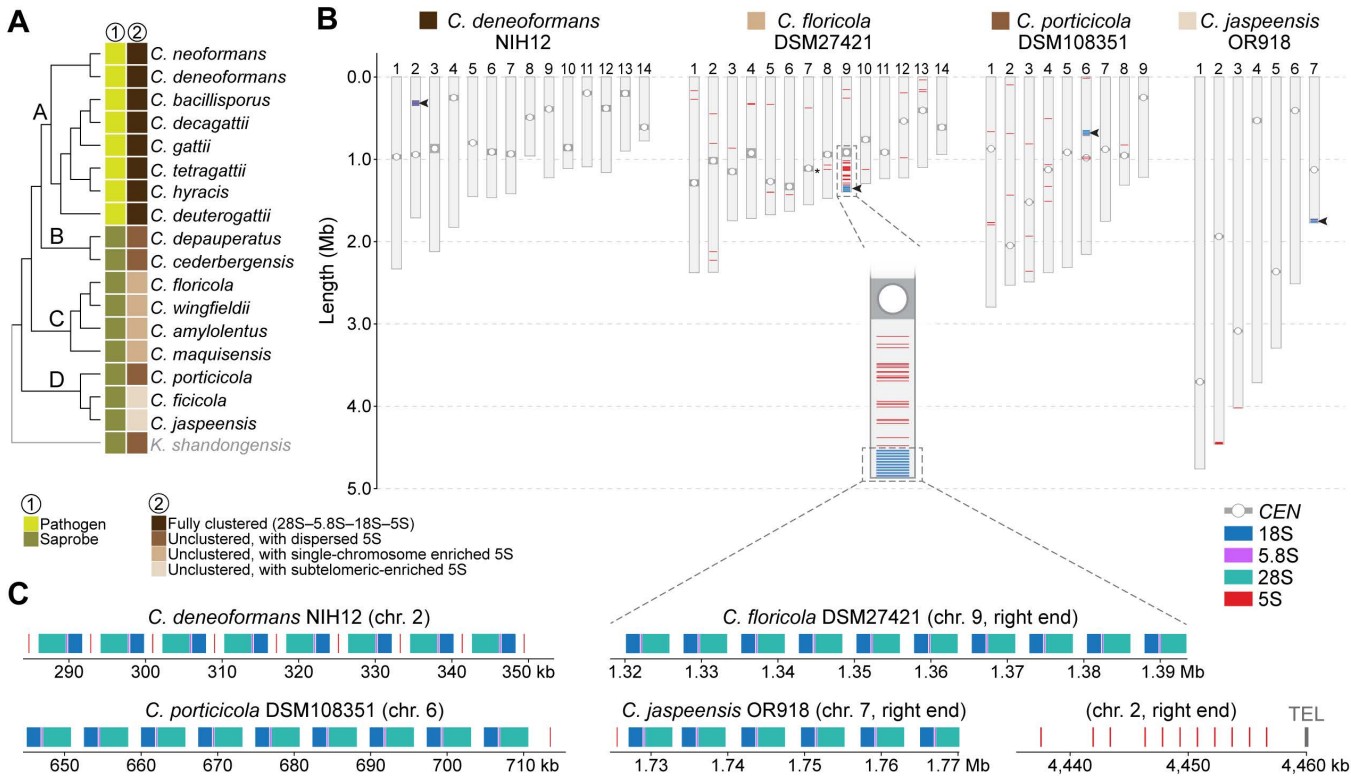

**Fig 6. Variation in ribosomal RNA (rRNA) gene organization across _Cryptococcus_ species.** (A) Cladogram of _Cryptococcus_ species (adapted from Fig 1), annotated with rRNA gene organization types. All pathogenic species from clade A exhibit a fully clustered rRNA array (28S–5.8S–18S–5S). In contrast, saprobic species display unclustered configurations with lineage-specific patterns: dispersed 5S genes (e.g., clade B), 5S enrichment on a single chromosome (e.g., clade C), or subtelomeric 5S enrichment (e.g., _C. ficicola_, _C. jaspeensis_ in clade D). (B) Genome-wide distribution of rRNA genes (28S, 5.8S, 18S, 5S) across four representative species: _C. deneoformans_ (clade A), _C. floricola_ (clade C), _C. porticicola_, and _C. jaspeensis_ (both clade D). Arrowheads indicate the position of the core rDNA array, and predicted centromeres are marked with open circles. In clade C, 5S genes are significantly enriched on a single chromosome (e.g., chr. 10 in _C. floricola_, see inset), whereas in clade D species, 5S genes are predominantly subtelomeric (e.g., chr. 2 in _C. jaspeensis_). See S7 Fig for additional species and strains. (C) Zoomed views of representative rDNA clusters and subtelomeric 5S regions. TEL, telomere.

To further explore clade-specific metabolic differentiation, we applied a Random Forest classifier to the binarized Biolog dataset. This analysis identified a ranked set of informative features, with glycerol, glucuronamide, L-serine, xylitol, and D-galacturonic acid among the top substrates contributing to clade discrimination (S8A Fig). Visualization of their average presence across clades revealed clear clade-specific patterns (S8B Fig). For example, glycerol and xylitol were utilized by all clades except clade A; glucuronamide utilization was largely restricted to clade A; and clades B and C showed impaired growth in 4% and 8% NaCl, consistent with heightened osmotic stress sensitivity (see also Taxonomy section). Conversely, D-melibiose utilization was restricted to clade C species. These trends suggest that substrate utilization profiles may reflect underlying phylogenetic structure and offer phenotypic markers for clade discrimination. To ensure reliability, a subset of Biolog results was cross-validated against standard assimilation tests from classical yeast taxonomy and physiological databases (see Methods for details). As an additional validation of these findings, we focused on glycerol and xylitol, the two most informative polyols identified by the Random Forest analysis, and performed spot growth assays on defined minimal media. The results confirmed the Biolog patterns: pathogenic clade A species exhibited little or no growth when glycerol or xylitol were provided as the sole carbon sources, whereas non-pathogenic species (clades B–D) grew robustly under the same conditions (S9 Fig).

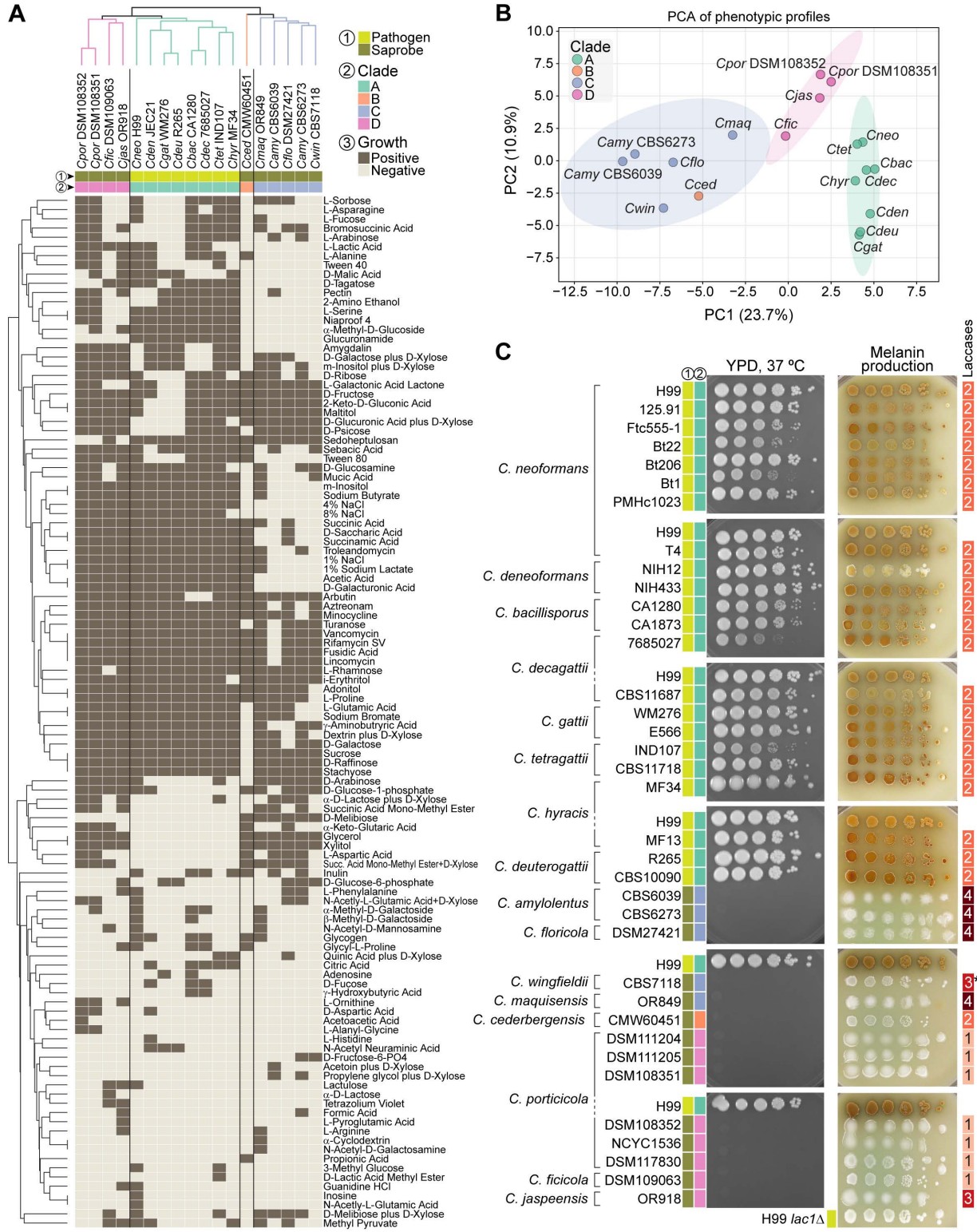

**Fig 7. Phenotypic profiling distinguishes *Cryptococcus* clades and pathogenic species.** (A) Clustermap showing hierarchical clustering of strains (columns) and informative phenotypic tests (rows), based on binary growth/no-growth responses in Biolog assays. Of the 153 substrates tested, 112 showed variable responses across strains and were retained for clustering. Growth was assessed using Biolog YT, FF, and GEN III MicroPlates. Binary

values indicate positive growth (brown) or absence of growth (beige). Strains are colored by phylogenetic clade, and clustering was performed using average linkage and Euclidean distance. Most strains cluster according to clade, with metabolic profiles broadly reflecting phylogenetic structure. (B) Principal component analysis (PCA) of Biolog growth profiles. The first two components explain 34.6% of the total variance. Ellipses represent 95% confidence intervals for each clade, based on a multivariate normal distribution. Most strains separate by clade, with the exception of *C. cederbergensis* (clade B), which clusters within the clade C ellipse, suggesting phenotypic similarity despite phylogenetic divergence. (C) Growth at 37°C (after 4 days) and melanin production on Bird seed agar (after 5 days at room temperature) across all tested strains. Strains were grown for 48 hours in YPD, washed, and spotted in 10-fold dilutions starting from an OD600 of 2.0. All strains from pathogenic species in clade A exhibited robust thermotolerance and melanin synthesis, whereas strains from saprobic species in clades B, C, and D showed neither trait. The rightmost column indicates the predicted number of laccase genes. In *C. wingfieldii*, one laccase-like gene is truncated and was therefore excluded from the count (marked with an asterisk). See S8C Fig for growth at other temperatures and results on alternative melanin-inducing medium (L-DOPA).

In parallel, we evaluated strains from all recognized *Cryptococcus* species (excluding *C. depauperatus*) for two traits broadly associated with pathogenicity: thermotolerance and melanin production. Growth at 37°C was tested on rich medium, and melanin synthesis was assessed on Bird seed agar and L-DOPA (Fig 7C and S8C). Thermotolerance was sharply restricted to clade A pathogens: all clade A strains grew robustly at 35°C and 37°C, while none of the clade B, C, or D strains did. Melanin production showed a similar distribution: all clade A strains produced some level of melanin on both media, whereas all nonpathogenic species lacked detectable production. This pattern was not explained by variation in laccase gene copy number. Laccases are oxidoreductase enzymes that catalyze melanin biosynthesis by oxidizing diphenolic compounds such as L-DOPA [54,55]. For instance, nonpathogenic clade C species such as *C. amylolentus* encode up to four predicted laccase genes but failed to produce melanin under the tested conditions indicating that gene copy number alone does not account for trait expression. Together, these results align with prior studies linking thermotolerance and melanin biosynthesis to virulence in *Cryptococcus* and suggest that these traits contribute to both ecological adaptation and the evolution of pathogenic potential.

## Environmental occurrence and limits of ITS-based detection of newly described species

Having established the genomic and phenotypic distinctness of the newly delineated *Cryptococcus* species, we next investigated their presence in environmental sequencing datasets to place them in a broader ecological context. Previous studies that uncovered some of these taxa reported their extremely rare occurrence in the environment [31,34]. As ITS-based metabarcoding is the most widely used approach in fungal biodiversity surveys [56], it holds promise for detecting rare fungi, including *Cryptococcus*. The broad availability of ITS metabarcoding data also offers an opportunity to assess how effectively current barcoding methods detect and identify these species in nature.

To this end, we performed exact-match searches of ITS1 and ITS2 sequences against the GlobalFungi database, which compiles over 80,000 fungal metabarcoding samples from diverse ecosystems [57]. Matches were considered only when query sequences were identical in both length and nucleotide composition. *Cryptococcus porticicola* was detected in nine samples from Europe, New Caledonia, and the United States, primarily from soil and decaying wood. Notably, four samples from northern Europe (Estonia, Finland, and Russia) were from coniferous forests dominated by Norway spruce (*Picea abies*), the typical habitat of *Ips typographus* (European spruce bark beetle), the main source of *C. porticicola* (S1 and S5 Tables). It was particularly abundant in dead bark samples from the Karelia region [58]. *Cryptococcus maquisensis* was detected in two samples from young *Populus deltoides* xylem collected in Knoxville, Tennessee (USA) [59]. No exact matches were found for *C. cederbergensis*, possibly suggesting a more restricted distribution.

While *C. cederbergensis*, *C. maquisensis*, and *C. porticicola* possess distinct ITS1 and ITS2 sequences, other species exhibit more limited marker resolution. For example, *C. ficicola* and *C. jaspeensis* differ in ITS1 but share an identical ITS2 sequence. ITS1 of *C. ficicola* matched a single air sample from Italy [60], while ITS1 of *C. jaspeensis* returned no matches. The shared ITS2 sequence was recovered from 20 soil samples in Italy, Switzerland [61,62], and New Caledonia [63]. Similarly, *C. hyracis* shares ITS1 identity with *C. tetragattii*, *C. deuterogattii*, and *C. bacillisporus*, and ITS2

identity with *C. deneoformans*, precluding species-specific identification. It should be noted that the GlobalFungi database is limited in insect-derived samples and lacks bark beetles entirely, providing little insight into insect-associated lineages. A full summary of GlobalFungi matches and ITS1/ITS2 resolution is provided in S5 Table, and an interactive map of matching samples is available as S2 File.

## Taxonomic description of novel *Cryptococcus* species

**Description of *Cryptococcus hyracis* M.C. Fisher, M. Vanhove, R.A. Farrer, M. David-Palma, M.A. Coelho, and J. Heitman sp. nov. (Index Fungorum: IF 904433).** Etymology: from hyrax (*Dendrohyrax arboreus*), an herbivorous mammal native to Africa and the Middle East; the species name refers to hyrax-associated environments, as several strains, including the type, were isolated from habitats such as tree holes and middens linked to hyrax activity.

After 1 week at 25°C on PD and GPY streak is white to cream, mucoid with a smooth glistening surface. Margins are smooth and entire. Cells are globose to ovoid 3–8 μm is size, occurring singly or in pairs, and proliferating by multilateral budding. Cells are surrounded by polysaccharide capsules. Pseudohyphae and true hyphae were not observed. Ballistoconidia were not observed. Teleomorph was not observed, as currently available strains are of the same mating type (*MAT*α).

Glucose is not fermented. Positive growth on D-Glucose, D-Galactose, L-Sorbose, D-Xylose, L-Arabinose, L-Rhamnose, Sucrose, Maltose, α,α-Trehalose, Me α-D-Glucoside, Cellobiose, Salicin, Arbutin splitting, Raffinose, Melezitose, Inulin, Starch, Ribitol, Arabinitol, D-Glucitol, D-Mannitol, Galactitol, myo-Inositol, 2-Keto-D-Gluconate, D-Gluconate, D-Glucuronate, D-Galacturonate, Succinate, Citrate, Ethanol, D-Glucarate, Galactonic acid Lactone, Palatinose, L-Malic acid, L-Tartaric acid, Galactaric acid, Gentiobiose, and Tween 80. No growth on D-Glucosamine, D-Ribose, D-Arabinose, Melibiose, Lactose, Glycerol, Erythritol, Xylitol, DL-Lactate, Methanol, Propane- 1,2 diol, Quinic acid, D-Tartaric acid, and Tween 40. No growth in the presence of 10% sodium chloride, but on 50% and 60% D-Glucose. Urea hydrolysis and Diazonium Blue B reaction are positive. Starch-like compounds are produced. Maximum growth temperature: 37°C.

Type: holotype, strain MF34, isolated from a tree hole associated with the Southern tree hyrax (*Dendrohyrax arboreus*), collected on 20 September 2013 in Mutinondo (latitude −12.45, longitude 31.29), Central Zambezian Miombo Woodlands; preserved in a metabolically inactive state in the German Collection of Microorganisms and Cell Cultures (DSMZ), Braunschweig, Germany as DSM 117690.

Molecular characteristics (holotype): nucleotide sequences of ITS and LSU (D1/D2 domains) are available under GenBank accession number PV828371.

**Description of *Cryptococcus cederbergensis* S. Marincowitz, N.Q. Pham, M. David-Palma, M.A. Coelho, J. Heitman, and M.J. Wingfield, sp. nov (Index Fungorum: IF 904434)**

Etymology: refers to the Cederberg Mountains in South Africa, where the sample that yielded the type strain was collected. Formed as a Latinized geographic adjective.

After 1 week at 25°C on PD and GPY agars, streak is white to cream, mucoid with a smooth glistering surface. On malt extract agar, culture is more mucoid. Margins are smooth and entire. Cells are globose to ovoid 3–8 μm is size, occurring singly or in pairs, and proliferating by multilateral budding. Cells are surrounded by polysaccharide capsules. Pseudohyphae and true hyphae were not observed. Ballistoconidia were not observed. Teleomorph was not observed.

Glucose is not fermented. Positive growth on D-Glucose, D-Galactose, D-Glucosamine, D-Ribose, D-Xylose, L-Arabinose, D-Arabinose, L-Rhamnose, Sucrose, Maltose, α,α-Trehalose, Me α-D-Glucoside, Cellobiose, Salicin, Arbutin splitting, Melibiose, Lactose, Raffinose, Melezitose, Inulin, Starch, Glycerol, Erythritol, Ribitol, Xylitol, Arabinitol, D-Glucitol, D-Mannitol, Galactitol, myo-Inositol, 2-Keto-D-Gluconate, D-Gluconate, D-Glucuronate, Succinate, Citrate, Methanol, Ethanol, D-Glucarate, Galactonic acid Lactone, Palatinose, L-Malic acid, Galactaric acid, Gentiobiose, and Tween 80. No growth on L-Sorbose, D-Galacturonate, DL-Lactate, Propane-1,2-diol, Quinic acid, L-Tartaric acid,

D-Tartaric acid, and Tween 40. No growth in the presence of 10% sodium chloride, but not on 50% D-Glucose. Urea hydrolysis and Diazonium Blue B reaction are positive. Starch-like compounds are produced. Maximum growth temperature: 30°C (weak, slow).

Type: holotype, dried specimen PRU(M) 4637, isolated from a bark beetle (*Lanurgus* sp.) infesting twigs of the endangered conifer *Widdringtonia cedarbergensis*, collected on 17 October 2022 in the Cederberg Mountains of South Africa, preserved in the H.G.W.L. Schweickerdt herbarium (PRU). Ex-holotype CMW 60451, preserved in a metabolically inactive state in the culture collection of the Forestry and Agricultural Biotechnology Institute (FABI) at the University of Pretoria, South Africa, as CMW-IA 7236, and in the German Collection of Microorganisms and Cell Cultures (DSMZ), Braunschweig, Germany, as DSM 117695.

Molecular characteristics (holotype): nucleotide sequences of ITS and LSU (D1/D2 domains) are available under GenBank accession number PV828378.

## Description of *Cryptococcus maquisensis* M.A. Coelho, M. David-Palma, O. Röhl, J.P. Sampaio, J. Heitman, and A.M. Yurkov, sp. nov. (Index Fungorum: IF 904435)

Etymology: from *maquis*, the French term for dense Mediterranean shrubland, referring to the habitat from which the species was isolated.

After 1 week at 25°C on PD and GPY agars, streak is white to cream, butyrous to mucoid with a smooth surface. Margins are smooth and entire. Cells are globose to ovoid 3–8 µm is size, occurring singly or in pairs, and proliferating by multilateral budding. Cells are surrounded by polysaccharide capsules. Short pseudohyphae observed in old cultures. No true hyphae were observed. Ballistoconidia were not observed. Teleomorph was not observed.

Glucose is not fermented. Positive growth on D-Glucose, D-Galactose, D-Glucosamine, D-Ribose, D-Xylose, D-Arabinose, L-Rhamnose, Sucrose, Maltose, α,α-Trehalose, Cellobiose, Salicin, Arbutin splitting, Melibiose, Raffinose, Melezitose, Starch, Glycerol, Erythritol, Ribitol, Xylitol, Arabinitol, D-Glucitol, D-Mannitol, Galactitol, myo-Inositol, 2-Keto-D-Gluconate, D-Gluconate, D-Glucuronate, Succinate, Citrate, Ethanol, D-Glucarate, Galactonic acid Lactone, Palatinose, L-Malic acid, L-Tartaric acid, Galactaric acid, and Gentiobiose. No growth on L-Sorbose, L-Arabinose, Me α-D-Glucoside, Lactose, Inulin, D-Galacturonate, DL-Lactate, Methanol, Propane-1,2-diol, Quinic acid, D-Tartaric acid, Tween 40, and Tween 80. Growth in the presence of 10% sodium chloride (weak), and on 50% and 60% D-Glucose. Urea hydrolysis and Diazonium Blue B reaction are positive. Starch-like compounds are produced. Maximum growth temperature: 30°C.

Type: strain OR849, isolated from soil, collected in September 2013 in Serra da Arrábida, Setúbal, Portugal; preserved in a metabolically inactive state in the Portuguese Yeast Culture Collection (PYCC), Caparica, Portugal (PYCC 10407, holotype), and in the German Collection of Microorganisms and Cell Cultures (DSMZ), Braunschweig, Germany (DSM 104548, isotype).

Molecular characteristics (holotype): nucleotide sequences of ITS and LSU (D1/D2 domains) are available under GenBank accession number PV828377.

## Description of *Cryptococcus ficicola* B. Turchetti, M. David-Palma, M.A. Coelho, A.M. Yurkov, and J. Heitman, sp. nov. (Index Fungorum: IF 904436)

Etymology: from Latin *ficus* (fig), referring to the fig fruit, the source of isolation.

After 1 week at 25°C on PD and GPY agars, streak is white to cream, butyrous to mucoid with a smooth surface. Margins are smooth and entire. Cells are globose to ovoid 3–8 µm is size, occurring singly or in pairs, and proliferating by multilateral budding. Cells are surrounded by polysaccharide capsules. Pseudohyphae and true hyphae were not observed. Ballistoconidia were not observed. Teleomorph was not observed.

Glucose is not fermented. Positive growth on D-Glucose, D-Galactose, D-Ribose, D-Xylose, D-Arabinose, L-Rhamnose, Sucrose, Maltose, α,α-Trehalose, Cellobiose, Salicin, Arbutin splitting, Raffinose, Melezitose, Starch, Glycerol, Erythritol, Ribitol, Xylitol, Arabinitol, D-Glucitol, D-Mannitol, Galactitol, myo-Inositol, 2-Keto-D-Gluconate,

D-Gluconate, D-Glucuronate, D-Galacturonate, Succinate, Ethanol, Palatinose, L-Malic acid, Gentiobiose, and Tween 80. No growth on L-Sorbose, D-Glucosamine, L-Arabinose, Me α-D-Glucoside, Melibiose, Lactose, Inulin, DL-Lactate, Citrate, Methanol, Propane-1,2-diol, Quinic acid, D-Glucarate, Galactonic acid Lactone, L-Tartaric acid, D-Tartaric acid, Galactaric acid, and Tween 40. Utilization of nitrogen sources: positive growth on potassium nitrate, sodium nitrite, ethylamine, lysine, and cadaverine. No growth in the presence of 10% sodium chloride (weak), and on 50% and 60% D-Glucose. Urea hydrolysis and Diazonium Blue B reaction are positive. Starch-like compounds are produced. Maximum growth temperature: 30°C.

Type: holotype, DBVPG 10121, isolated from figs collected in September 2011 in Brufa, Perugia, Italy, preserved in a metabolically inactive state in the Industrial Yeasts Collection DBVPG, Department of Agricultural, Food and Environmental Sciences, University of Perugia, Perugia, Italy. Ex-holotype is preserved in a metabolically inactive state in the German Collection of Microorganisms and Cell Cultures (DSMZ), Braunschweig, Germany as DSM 109063.

Molecular characteristics (holotype): nucleotide sequences of ITS and LSU (D1/D2 domains) are available under GenBank accession number PV828380.

### Description of *Cryptococcus jaspeensis* M.A. Coelho, M. David-Palma, O. Röhl, J.P. Sampaio, J. Heitman, and A.M. Yurkov, sp. nov. (Index Fungorum: IF 904437)

Etymology: from Alto do Jaspe, a locality in the Arrábida Natural Park (Portugal) where the type strain was isolated. The epithet *jaspeensis* is a Latinized toponym referring to the place of origin.

After 1 week at 25°C on PD and GPY agars, streak is white to cream, butyrous to mucoid with a smooth surface. Margins are smooth and entire. Cells are globose to ovoid 3–8 μm is size, occurring singly or in pairs, and proliferating by multilateral budding. Cells are surrounded by polysaccharide capsules. Pseudohyphae and true hyphae were not observed. Ballistoconidia were not observed. Teleomorph was not observed.

Glucose is not fermented. Positive growth on D-Glucose, D-Galactose, D-Glucosamine, D-Ribose, D-Xylose, D-Arabinose, L-Rhamnose, Sucrose, Maltose, α,α-Trehalose, Me α-D-Glucoside, Cellobiose, Salicin, Arbutin splitting, Raffinose, Melezitose, Inulin, Starch, Glycerol, Erythritol, Ribitol, Xylitol, Arabinitol, D-Glucitol, D-Mannitol, Galactitol, myo-Inositol, 2-Keto-D-Gluconate, D-Gluconate, D-Glucuronate, D-Galacturonate, Succinate, Ethanol, D-glucarate, Palatinose, L-Malic acid, Galactaric acid, Gentiobiose, Tween 40, and Tween 80. No growth on L-Sorbose, L-Arabinose, Melibiose, Lactose, DL-Lactate, Citrate, Methanol, Propane-1,2-diol, Quinic acid, Galactonic acid Lactone, L-Tartaric acid, and D-Tartaric acid. Utilization of nitrogen sources: positive growth on potassium nitrate, sodium nitrite, ethylamine, and lysine. No growth in the presence of 10% sodium chloride (weak), and on 50% and 60% D-Glucose. Urea hydrolysis and Diazonium Blue B reaction are positive. Starch-like compounds are produced. Maximum growth temperature: 25°C.

Type: strain OR918, isolated from soil collected in September 2013 in Serra da Arrábida, Setúbal, Portugal, preserved in a metabolically inactive state in the Portuguese Yeast Culture Collection (PYCC), Caparica, Portugal (PYCC 10408, Holotype), and in the German Collection of Microorganisms and Cell Cultures (DSMZ), Braunschweig, Germany (DSM 104549, isotype).

Molecular characteristics (holotype): nucleotide sequences of ITS and LSU (D1/D2 domains) are available under GenBank accession number PV828379.

### Description of *Cryptococcus porticicola* M.A. Coelho, M. David-Palma, A.V. Kachalkin, T.A. Kuznetsova, M. Kolarik, A.M. Yurkov, and J. Heitman, sp. nov. (Index Fungorum: IF 904438)

Etymology: from Latin *porticus* (meaning gallery, passage, or corridor) referring to the burrows or tunnels made by bark beetles, which is the substrate where the species has been repeatedly found.

After 1 week at 25°C on PD and GPY agars, streak is white to cream, butyrous to mucoid with a smooth surface. Margins are smooth and entire. Cells are globose to ovoid 3–8 μm is size, occurring singly or in pairs, and proliferating by multilateral budding. Cells are surrounded by polysaccharide capsules. Pseudohyphae and true hyphae were not observed

under these conditions. Ballistoconidia were not observed. Under mating conditions on V8 agar (pH 5.0), hyphae and basidia-like structures were sporadically observed after ~1 month of incubation in co-culture with strain NCYC1536. These putative sexual structures appeared embedded in the agar, forming small clusters of globose cells approximately 8–10 µm in diameter. Their morphology was consistent with basidia, but their identity could not be definitively confirmed. No sexual structures were observed on MS medium.

Glucose is not fermented. Positive growth on D-Glucose, D-Galactose, L-Sorbose, D-Glucosamine, D-Ribose, D-Xylose, L-Arabinose, D-Arabinose, L-Rhamnose, Sucrose, Maltose, α,α-Trehalose, Me α-D-Glucoside, Cellobiose, Salicin, Arbutin splitting, Raffinose, Melezitose, Starch, Glycerol, Erythritol, Ribitol, Xylitol, Arabinitol, D-Glucitol, D-Mannitol, Galactitol, myo-Inositol, 2-Keto-D-Gluconate, D-Gluconate, D-Glucuronate, D-Galacturonate, Succinate, Ethanol, D-Glucarate, Galactonic acid Lactone, Palatinose, L-Malic acid, Galactaric acid, Gentiobiose, Tween 40, and Tween 80. No growth on Melibiose, Lactose, Inulin, DL-Lactate, Citrate, Methanol, Propane-1,2-diol, Quinic acid, L-Tartaric acid, and D-Tartaric acid. Utilization of nitrogen sources: positive growth on potassium nitrate, sodium nitrite, ethylamine, and lysine. No growth in the presence of 10% sodium chloride (weak), and on 50% and 60%D-Glucose. Urea hydrolysis and Diazonium Blue B reaction are positive. Starch-like compounds are produced. Maximum growth temperature: 30°C.

Type: from gallery of great spruce bark beetle, *Dendroctonus micans* collected in May 2017 in the vicinity of Zvenigorod town, Moscow region, Russia, preserved as two metabolically inactive isotypes: in the yeast collection of the Lomonosov Moscow State University (KBP Y-6209, holotype); and the All-Russian Collection of Microorganisms (VKM), Pushchino, Russia (VKM Y-3025, isotype). Ex-type culture is preserved in a metabolically inactive state in the German Collection of Microorganisms and Cell Cultures (DSMZ), Braunschweig, Germany as DSM 108351.

Paratypes: KBP Y-6300 (= DSM 108352 = VKM Y-3027) from the gallery of great spruce bark beetle, *Dendroctonus micans* (in the vicinity of Dmitrov town, Moscow region, Russia), KBP Y-6370 (= DSM 111204 = VKM Y-3517) and KBP Y-6556 (= DSM 111205 = VKM Y-3589) from spruce bark beetle gut, *Ips typographus* (in the vicinities of Nakipelovo and Gzhel villages, Moscow region, Russia), CCUG 11125 (= NCYC 1536 = DSM 117691) from spruce bark beetle gut, *Ips typographus* (Göteborg, Sweden), and CCF 6641 (= DSM 117830) from spruce bark beetle gut, *Ips typographus* (Nižbor, Czechia).

Molecular characteristics (holotype): nucleotide sequences of ITS and LSU (D1/D2 domains) are available under GenBank accession number PV828382.

## Discussion

This study expands the known diversity of the genus *Cryptococcus* by formally describing six new species distributed across four distinct clades. Although initially identified as potential new *Cryptococcus* species through rRNA and later genome sequencing, these taxa had remained undescribed, limiting their integration into broader ecological and evolutionary contexts. Defining species boundaries in fungi, especially among closely related lineages, typically requires a holistic approach that integrates phylogenetic relationships, evidence of reproductive isolation, and other biological traits. However, such data are often difficult to obtain, particularly for non-model organisms with limited experimental tractability or for lineages represented by few isolates. In these cases, genomic data offer a useful alternative, though interpretation is not without challenges.

As fungal genome resources continue to grow, fungal taxonomy is entering a phylogenomic era in which genome-scale data and quantitative divergence metrics provide a rapid, reproducible, and evolutionarily grounded basis for species delineation [36,38]. In this context, taxogenomic approaches anchored in genome-wide comparisons and robust phylogenetic inference are particularly valuable for uncovering hidden diversity and resolving taxonomic uncertainty in yeasts [64,65]. Still, metrics such as ANI and AAI must be benchmarked against external data reflecting fungal biology, including ecology and reproduction—traits traditionally employed for species circumscription under the phenotypic and biological species concepts [56,66].

Species boundaries in this study were defined primarily by genome-scale similarity patterns. Lineages were recognized as distinct species when they formed well-supported monophyletic groups and showed genome-wide similarity lower than the minimum observed among conspecifics of the well-circumscribed reference species *C. neoformans*, that is, below the maximum genomic divergence still consistent with being the same species. This within-species range provided an empirical upper bound for intraspecific similarity, which was contrasted with the interspecific similarity observed between the closely related species *C. bacillisporus* and *C. decagattii*, representing the highest ANI values observed between distinct species. Lineages with similarity values falling below this interspecific range were therefore interpreted as distinct species, whereas those with intermediate values may represent borderline or incipient cases. Because absolute genomic thresholds for fungal species remain undefined, these criteria were applied empirically and interpreted within a comparative rather than prescriptive framework. Additional evidence, such as differences in chromosomal structure, sexual compatibility, or ecology, provided supporting context where available. When different datasets were not fully congruent, genome-wide similarity and phylogenetic distinctness were prioritized as the most stable indicators of long-term evolutionary independence, whereas reproductive, structural, and phenotypic traits (with more limited coverage) were considered complementary and often asynchronous indicators of speciation. This integrative framework ensured consistency even when experimental assessments of mating compatibility or progeny viability were limited or variable among strains. Although applied here to *Cryptococcus*, this approach can be extended to other basidiomycete genera, particularly those that can be benchmarked against existing mating-compatibility tests and/or documented host-range data.

In our phylogenomic analyses, residual conflict remained at both deep and shallow nodes, although strong overall concordance was observed across inference methods. For instance, while the placement of clades B and C relative to clade A was topologically consistent, reduced gene and site concordance and lower quartet support at this deep node point to lingering uncertainty, likely reflecting ancient incomplete lineage sorting (ILS). Within clade A, several internal nodes also exhibited gene tree discordance, likely due to a combination of ILS and recent, rapid speciation. Although historical introgression cannot be entirely ruled out within the *Cryptococcus gattii* complex, available evidence suggests it is rare and limited in scope [19,67]. Notably, coalescent-based inference with ASTRAL yielded higher resolution across all clade A nodes, with quartet support values exceeding the 33% threshold. These patterns suggest that gene tree discordance, while common in recently diverged fungal lineages, does not necessarily compromise species tree inference, particularly when relationships are reconstructed with coalescent-aware approaches like ASTRAL, which explicitly model the effects of ILS.

Genome-wide similarity metrics such as ANI and dDDH provide a complementary and quantitative framework for evaluating species boundaries, as demonstrated in prokaryotes [37]. Although fungi lack a universally accepted genomic threshold for species delimitation, a 95% ANI cutoff has been proposed based on limited comparisons across a few selected ascomycetous and basidiomycetous yeasts. However, our comparative analyses support emerging evidence that this threshold may not be suitable and underestimate species-level divergence in some basidiomycete lineages [38,68]. For example, the lowest ANI and dDDH values observed among *C. neoformans* strains (97.7% and 79%, respectively) provide a conservative lower bound for intraspecific genomic similarity, while the highest ANI and dDDH values between *C. bacillisporus* and *C. decagattii* (96.5% ANI; 69.4% dDDH), two recognized species, establish a practical upper bound for interspecific genomic similarity.

The newly described species *C. hyracis* sp. nov. illustrates the utility of this framework. Phylogenomic analyses resolved *C. hyracis* as sister to *C. tetragattii*, with ANI values ranging from 95.32% to 95.51%, and dDDH values from 62.1% to 62.8%. These values fall below both the intraspecific thresholds observed in *C. neoformans* and even the interspecific values seen between *C. bacillisporus* and *C. decagattii*, providing quantitative support for species-level distinction. Although all currently available *C. hyracis* strains are *MAT*α, precluding intraspecific mating assays, interspecific crosses with *MAT***a** strains from other species produced sexual structures, suggesting the lineage retains sexual competence. Further studies evaluating progeny viability and reproductive compatibility with *C. tetragattii* are needed but are constrained

by the current lack of *C. hyracis MAT***a** isolates for control crosses. More broadly, systematic reproductive assessments across the *C. gattii* complex would help refine the interpretive power of genome-based divergence thresholds. When standardized and applied in a clade-specific context, these genomic similarity metrics should provide a scalable and reproducible framework for species delimitation, particularly in lineages where experimental crosses are infeasible. These comparative benchmarks will be refined as broader sampling, including opposite mating types, and additional genome sequencing improve clade-specific calibration. The presence of hybrid strains remains a caveat, underscoring the need to integrate genomic evidence with other lines of data. Together, these findings extend earlier support for the distinctiveness of *C. hyracis* [19] and reinforce the broader utility of empirical, lineage-specific benchmarks for taxonomic resolution in *Cryptococcus* and other fungal taxa.

Chromosomal architecture and genome-wide sequence divergence do not always evolve in concert, and both can independently shape species boundaries. For instance, *C. hyracis* and *C. tetragattii* are resolved as sister species and differ by ~4.5% at the genome level, yet their karyotypes are nearly collinear apart from a single large inversion on chromosome 2, which would only impact fertility in the rare case of crossover within the inverted segment. A similar decoupling of sequence and structural evolution is observed between *C. neoformans* 125.91 and *C. deneoformans* NIH433, which diverge by ~12% but differ structurally by only two large-scale inversions (on chromosomes 1 and 9). This pattern echoes observations in other fungal systems. In *Saccharomyces*, for instance, *S. cerevisiae* and *S. paradoxus* retain collinear genomes but produce sterile hybrids due to anti-recombination effects mediated by mismatch repair [69,70]. In *Cryptococcus*, deletion of *MSH2* (a key component of the mismatch repair system) enhances hybrid viability between *C. neoformans* and *C. deneoformans*, yet fails to restore normal levels of meiotic recombination, implying the presence of additional post-zygotic barriers [71]. Adding further nuance, some *C. neoformans* strains (e.g., H99, Bt65, Bt81) harbor private rearrangements [43,72], demonstrating that chromosomal changes can arise independently within species and may not reflect broader lineage evolution. In contrast, *C. bacillisporus* and *C. decagattii*, the most closely related species pair in our dataset (~3.5% divergence), exhibit up to two fixed reciprocal translocations between their sequenced strains, despite high genome-wide similarity. Whether these rearrangements initiated speciation or arose secondarily remains unresolved. However, experimental work in *C. neoformans* shows that targeted centromere breakage can generate novel karyotypes that reduce meiotic viability in the absence of sequence divergence, supporting a causal role for chromosomal changes in reproductive isolation [43]. Although often underappreciated, these findings draw further attention to the role of chromosomal rearrangements in fungal speciation and illustrate the importance of broad within-species sampling to fully capture karyotypic dynamics and evolutionary trajectories.

In addition to the structural rearrangements observed among closely related pathogenic species, our analyses revealed more extensive karyotypic shifts in clade D, including multiple reductions in chromosome number. The three newly described clade D species (*C. porticicola*, *C. ficicola*, and *C. jaspeensis*) harbor fewer chromosomes (9, 8, and 7, respectively) than the ancestral 14-chromosome karyotype inferred for *Cryptococcus* and its sister genus *Kwoniella* [32]. Comparative synteny analyses of centromeric regions suggest that these chromosome number reductions were primarily driven by structural rearrangements followed by centromere inactivation. Several centromere-syntenic regions retain conserved flanking gene order but lack TE enrichment and display only residual 5mCG, consistent with inactivated centromeres (*iCENs*). This aligns with broader models in which centromere identity erodes through the progressive loss or excision of repeat-rich DNA, as reported in *C. depauperatus* [32] and other fungi such as *Verticillium*, *Malassezia* and *Candida* [52,53,73,74], supporting centromere inactivation as a recurring process in karyotype evolution. We also identified an intercentromeric recombination (ICR) event in *C. porticicola* strain DSM111204, associated with a strain-specific chromosomal rearrangement. ICR has been reported in both pathogenic and nonpathogenic *Cryptococcus* species [32,43,45], and its detection here extends the known phylogenetic distribution of this mechanism.

While centromere inactivation and ICR can explain many of the observed rearrangements, the patterns in *C. ficicola* and *C. jaspeensis* point to an additional process. In these species, the total number of inferred *iCENs* surpasses the

number of chromosomes lost when compared to *C. porticicola*, suggesting that centromere inactivation alone does not fully account for their current karyotypes. This raises the possibility that centromere repositioning or neocentromere formation may have accompanied or followed inactivation events to maintain chromosome stability. A candidate centromere shared by both species, located on chromosomes 4 and 5, respectively, exhibits hallmark centromeric features, including enrichment for TEs and 5mCG DNA methylation. The syntenic region in *C. porticicola* shows only partial enrichment for these features, suggesting that it may represent either a newly co-opted centromere in the ancestor of *C. ficicola* and *C. jaspeensis*, or an ancestral centromere that was subsequently lost or inactivated in *C. porticicola*. If this region indeed functions as a centromere, its architecture contrasts with neocentromeres described in *C. deuterogattii*, which were experimentally induced by targeted deletion of native centromeres and typically form at gene-rich, repeat-poor, and unmethylated loci and are often epigenetically unstable [50,51]. This distinction may reflect either species-specific differences or contrasts between newly formed neocentromeres and those that have stabilized over extended evolutionary periods. Future work will aim to functionally test centromere activity once these species become genetically tractable, which will enable targeted validation (e.g., via CENP-A ChIP-seq) and experimental assessment of centromere dynamics. Taken together, our findings suggest that centromere inactivation, intercentromeric recombination, and centromere repositioning are recurrent processes contributing to genome restructuring and karyotype evolution in *Cryptococcus*, with potential implications for lineage diversification and speciation.

Ribosomal RNA (rRNA) genes have long served as core phylogenetic and taxonomic markers in fungal systematics [56,66]. Mirroring the broader chromosomal and centromeric changes observed across *Cryptococcus*, we found lineage-specific variation in the genomic arrangement of the 5S rRNA gene across *Cryptococcus* species. In clade A pathogens, 5S genes remain physically clustered with the 18S, 5.8S, and 28S components on a single chromosome, a configuration common in fungi [75] and likely maintained by constraints related to nucleolar organization and coordinated rRNA dosage control [76]. In contrast, nonpathogenic species exhibit unlinked and frequently dispersed 5S loci, often enriched in subtelomeric regions as seen in clade D species such as *C. ficicola* and *C. jaspeensis*. While some clade C species retain partial enrichment of 5S genes on the same chromosome as the canonical rDNA array, others show evidence of array relocation, consistent with rDNA mobility. These patterns may reflect relaxed functional constraints or increased genome plasticity in saprobic lineages. Interestingly, in *Schizosaccharomyces octosporus* and *S. osmophilus*, rapidly evolving *wtf* poison–antidote genes are frequently positioned adjacent to the highly dynamic 5S rDNA repeats; this genomic proximity suggests that the repeat-rich 5S neighborhood may facilitate duplication and diversification of these selfish elements, accelerating local genome evolution [77]. While no equivalent meiotic drive system has been characterized in *Cryptococcus*, this pattern raises the possibility that analogous repeat-mediated processes could underlie the rapid 5S architecture changes observed in this genus. Mechanistically, the genomic decoupling of 5S from the other rRNA genes may be facilitated by its independent transcription by RNA polymerase III, in contrast to the polymerase I-transcribed 18S, 5.8S, and 28S genes. This transcriptional distinction has been proposed as a key reason for the physical separation of 5S genes in many eukaryotes [78]. Additionally, retroelement-mediated mobility has been hypothesized as a mechanism for 5S gene dispersal, based on shared internal promoters between 5S rRNA genes and short interspersed nuclear elements (SINEs), and documented interactions with long interspersed nuclear elements (LINE) machinery [78,79]. The subtelomeric bias observed in some *Cryptococcus* species is also consistent with known hotspots of recombination and TE activity [72]. Together, our results indicate that 5S rRNA gene organization is evolutionarily labile in *Cryptococcus*, likely influenced by its independent transcription and potential for mobilization. These lineage-specific differences raise important questions about the functional consequences of rDNA organization, including possible effects on nucleolar structure, rRNA transcriptional regulation, or genome stability. While these impacts remain unclear, the extent of observed divergence suggests that rRNA gene architecture represents an underappreciated axis of genome evolution in *Cryptococcus* and positions this genus as a promising model for investigating rRNA gene dynamics and genome compartmentalization in fungi.

Beyond structural genome evolution, we also explored the recombination potential of *C. porticicola*, the only newly described nonpathogenic species for which multiple isolates of opposite mating types were available. All six strains were isolated from bark beetle galleries or larval guts, indicating a strong ecological association with conifer-infesting beetles. In mating assays, only one cross (DSM108352×NCYC1536) produced hyphae and embedded basidia-like structures resembling those of *Kwoniella* rather than the typical aerial basidia of *Cryptococcus*, suggesting a potentially divergent mode of sexual development. Although these structures emerged only sporadically and their identity requires further ultrastructural or genetic confirmation, they appeared morphologically distinct from the chlamydospore-like cells reported in *C. neoformans* under stress or nutrient-limiting conditions, which are typically more darkly refractile and seem to possess thicker cell walls [80]. Notably, the two mating strains differ by two chromosomal translocations, which are likely to reduce progeny viability. Despite limited morphological evidence, genomic analyses based on a small number of isolates support ongoing sexual reproduction in *C. porticicola*. Genome-wide SNP patterns revealed reticulate relationships and measurable LD decay, with LD50 approximately 13.9 kb. This value is broadly comparable to those reported for recombining *C. neoformans* subpopulations (5–7.5 kb when analyzed separately; [33]) and to those observed in *S. paradoxus* (LD50~9 kb; [81]). Although the number of isolates remains limited, these comparable magnitudes indicate that recombination likely occurs in natural populations of *C. porticicola*, albeit potentially at lower effective frequency or within localized population contexts. While additional sampling will be needed to clarify the extent of sexual reproduction, the combined genomic and morphological evidence suggests that meiotic processes likely contribute to genetic cohesion, evolutionary flexibility, and ecological persistence in this species. Future work will focus on optimizing mating conditions to enable successful crosses between strains with more collinear karyotypes, as all other combinations tested to date have not yielded evidence of mating. In parallel, recovering additional strains from bark beetle-associated substrates may help identify more sexually compatible pairs. If these approaches prove unsuccessful, or if sporulation remains rare or progeny are technically difficult to isolate, introducing selectable markers into compatible strains could provide an alternative means to detect genetic exchange following co-culture, as demonstrated in other non-model *Cryptococcus* lineages [25].

Complementing our genomic analyses, we evaluated phenotypic variation across the *Cryptococcus* genus to determine whether clade-level divergence is mirrored by metabolic traits. Comparative growth profiling revealed clear clade-level differentiation: hierarchical clustering and principal component analysis (PCA) based on Biolog substrate utilization patterns, benchmarked against standard assimilation tests and culture-collection databases, showed that species within the same clade tend to group together phenotypically, broadly reflecting phylogenetic relationships. Notably, *C. cederbergensis* (clade B) clustered within the clade C phenotypic space, suggesting functional convergence potentially linked to similar environmental conditions. Despite the complexity of the experimental design and validation framework, the consistent clade-level growth responses and clustering observed support its utility for comparative phenotyping and highlight candidate traits for future genomic dissection. Under the assay conditions used here, glycerol and xylitol, two informative substrates identified by the Random Forest analysis, supported robust growth of saprobic species from clades B, C, and D, but only limited or no growth of pathogenic clade A species, when provided as sole carbon sources.

Xylitol originates from xylose, a major constituent of plant hemicellulose, and occurs at low levels in plant saps, fruits, and decaying wood, supporting its biogenic availability in plant-rich environments. In wood-decay and insect-associated microhabitats, xylose/xylitol turnover is common: multiple yeasts isolated from rotting wood actively ferment xylose and produce xylitol [82], and similar xylose-fermenting yeasts have been documented in the guts of wood-feeding beetles [83,84]. Soil yeasts likewise utilize intermediates of plant degradation [85], which may explain the isolation of *C. jaspeensis* and *C. maquisensis* from soils in Portugal [34]. Similar to xylitol, glycerol and related polyols occur in bark-beetle–associated habitats, where they seem to accumulate in insect tissues as cryoprotectants and metabolic intermediates [86]. The prevalence of xylitol- and glycerol-positive species among clades B, C, and D, many of which are bark- or insect-associated, may therefore reflect adaptive responses to polyol-enriched microhabitats. While preliminary, these observations provide testable hypotheses for future work connecting substrate use and occupied niche and motivate targeted

assays under ecologically relevant conditions. Similar to findings in *Saccharomycotina* yeasts, where ecological transitions often coincide with extensive remodeling of metabolic gene repertoires [87,88], our phenotype-first approach in *Cryptococcus* also establishes a foundational framework to connect metabolic traits with ecological strategies and historical adaptation. Future comparative genomic analyses will clarify whether these phenotypic shifts reflect underlying changes in gene content or regulation.

Our Biolog screens and validation assays were performed at 25°C for standardization, which may not reflect the optimal conditions for substrate utilization across species, nor account for potential metabolic activity under natural or host-associated environments. For instance, although growth in glycerol was recorded as negative in both our experiments and taxonomic literature, studies have reported glycerol utilization by *C. neoformans* strains at higher temperatures [89–91]. Thus, the apparent clade-level differences in glycerol/xylitol utilization should be interpreted as condition-dependent, and they raise the hypothesis that substrate use in pathogenic clade A species may shift at elevated temperatures relevant to host environments. Systematic testing across temperatures (e.g., 25–37°C), and under host-mimicking conditions, will be necessary to resolve these interactions.

Strikingly, we confirmed that two traits historically associated with virulence in *Cryptococcus*—thermotolerance and melanin production—are restricted to clade A pathogens in our dataset, under the tested conditions. Yet, in the case of melanin production, nonpathogenic species retain a full complement of known biosynthetic genes [32], including multiple laccase homologs; several even harbor greater copy numbers than their pathogenic counterparts. In *C. neoformans*, melanin synthesis is largely driven by the cell wall–associated laccase *LAC1*, which plays a central role in oxidative stress resistance and survival within macrophages [92,93], while its paralog *LAC2* can partially compensate under certain conditions [94]. The retention and potential redundancy of these laccase genes in nonpathogenic lineages therefore likely reflect ancestral ecological functions (such as oxidation of environmental phenolics and plant-derived compounds, or detoxification of reactive metabolites), that were later co-opted to support virulence in pathogenic species. This interpretation is consistent with models of exaptation, in which virulence-associated traits arise from preexisting adaptations to diverse ecological niches [95]. Beyond laccase, classical physiological work in *C. neoformans* has shown that melanization requires active, high-affinity uptake of exogenous catechol and catecholamine precursors (e.g., L-DOPA, dopamine, norepinephrine) [96], although the specific transporter(s) responsible remain unidentified. With respect to thermotolerance, its strict restriction to clade A pathogens reinforces its status as a critical prerequisite for infection and survival in mammalian hosts. Stressors encountered during infection, such as elevated temperature, have been shown to trigger transposable element mobilization in both *C. deneoformans* and *C. neoformans*, resulting in hypermutator-like phenotypes and accelerated antifungal resistance [97–99]. Although none of the nonpathogenic species analyzed here can grow at 37°C, several retain high TE burdens [32], raising the possibility that stress-induced genomic plasticity may also occur in environmental contexts and could, under the right conditions, prime lineages for future adaptation. Investigating whether similar mutational dynamics operate in these lineages could reveal a latent evolutionary capacity for pathogenicity. Further work integrating pan-genomic and pan-phenomic analyses, ecological data, and broader taxon sampling will be key to resolving how such traits shape adaptation and diversification across the *Cryptococcus* genus.

The ecological breadth of *Cryptococcus* is now more apparent with the recovery of newly described species from bark beetle galleries, fruit, and soil across diverse geographic regions. Although distinct from the arboreal hollows, avian guano, and forest debris typically associated with *C. neoformans* and *C. gattii* senso lato, these substrates also represent organic-rich, microbially dynamic environments that may support long-term persistence and saprobic growth. These findings demonstrate the importance of environmental sampling in revealing hidden diversity and advancing our understanding of the ecological and evolutionary processes shaping the genus. They also suggest that complex environmental niches may serve as incubators for traits involved in host adaptation and pathogenicity. In this context, the formal recognition of phylogenetically distinct lineages is not taxonomic excess but a necessary step toward cataloging fungal diversity, linking genotypes to phenotypes. While our integrated approach convincingly resolved species

boundaries, it also revealed a key challenge for future surveys: ribosomal markers such as ITS and LSU lack the resolution to distinguish closely related *Cryptococcus* species, limiting species-level identification in short-read metabarcoding datasets. This highlights the need for continued development of genome-informed frameworks to enable consistent identification of novel species and strains, which are essential to support long-term study, surveillance, and diagnostic development, especially as some species may emerge as clinically relevant under shifting ecological, climatic, or antifungal selective pressures.

## Materials and methods

### Strains, mating assays, and microscopy

Strains analyzed in this study were routinely cultured on YPD medium (10 g/L yeast extract, 20 g/L Bacto Peptone, 20 g/L dextrose, and 20 g/L agar), unless specified otherwise. *Cryptococcus neoformans*, *C. deneoformans*, and *C. gattii* sensu lato strains were incubated at 30°C, while other *Cryptococcus* species were maintained at room temperature (20–25°C). A complete list of strains and associated metadata is provided in S1 Table.

Mating assays were performed by mixing equal amounts of cells from each strain on V8 agar media (10 g/L yeast extract, 20 g/L Bacto Peptone, 20 g/L dextrose, 20 g/L agar; pH 5.0) or Murashige and Skoog (MS) agar media (4.328 g/L MS mix, 16 g/L agar, 1X vitamin mix, pH to 5.8) and incubated in the dark at room temperature (conditions known to induce mating in *C. neoformans*) for up to one month [100]. In the case of *C. porticicola*, incubation was extended due to delayed and sporadic hyphal development, which was first observed around the one-month mark. Plates were regularly monitored microscopically for the development of mating structures. Solo cultures of each strain were also evaluated on V8 pH 5 and/or MS without a compatible partner. Assays were repeated three independent times to ensure consistent phenotypic observations.

To assess hyphal growth, basidia, and spore formation, the edges of yeast colonies and mating patches were examined microscopically and photographed. Specifically for *C. porticicola*, putative sexual structures embedded within the mating media at the patch margins were imaged by excising small agar blocks and gently compressing them between a microscope slide and coverslip. Light microscopy was performed using a Zeiss Axio Scope.A1 microscope equipped with an Axiocam Color camera and ZEN Lite V3.4 software. Samples of sporulating colonies were prepared for scanning electron microscopy (SEM) as previously described [101] and imaged using the Apreo S SEM with an EDS detector (ThermoFisher, USA) at the Shared Materials Instrumentation Facility at Duke University.

### Genomic DNA extraction

High-molecular-weight (HMW) genomic DNA was isolated using a cetyltrimethylammonium bromide (CTAB)-based extraction method as previously described [43], with careful handling to reduce mechanical shearing. DNA purity was assessed with a NanoDrop spectrophotometer (Thermo) by measuring A260/A280 and A260/A230 ratios. DNA integrity and fragment length were evaluated by clamped homogeneous electric field (CHEF) electrophoresis [32]. For Illumina short-read whole-genome sequencing, genomic DNA was extracted using a phenol:chloroform-based procedure [32]. Final concentrations were quantified using the Qubit dsDNA Assay Kit (Invitrogen) on a Qubit fluorometer.

### Genome sequencing

Whole-genome sequencing was performed with Oxford Nanopore and Illumina technologies. Illumina sequencing for strains DSM111204, DSM111205 and DSM108352 was conducted at the Duke University Sequencing and Genomic Technologies (DUSGT) Core. Libraries were prepared with the Kapa HyperPlus kit and sequenced as paired-end 2 × 150 bp reads on Illumina Novaseq. Nanopore sequencing for strains DSM111204 and DSM108352 was performed in-house. Samples were barcoded using the SQK-LSK109 ligation sequencing kit in combination with EXP-NBD104 native barcoding kits. Library preparation followed the manufacturer's protocols, and sequencing was carried out on R9.4.1 flow cells

(FLO-MIN106) for 48–72 hours under default voltage on a MinION Mk1B device. The corresponding version of MinION software available at the time of each run was applied. Additional details on sequencing platforms, basecalling, and demultiplexing for each sample are provided in S6 Table.

**Genome assembly, gene prediction, and functional annotation**

Complete nuclear genomes of *C. porticicola* strains DSM111204 and DSM108352 were assembled using Canu [102] with Nanopore long-read data and polished with both Nanopore and Illumina reads, following a previously established workflow [30]. This process included iterative error correction, removal of mitochondrial and rDNA-only contigs, assessment of assembly completeness and telomeric regions, and manual inspection of read coverage using IGV. The draft genome assembly of *C. porticicola* strain DSM111205 was generated using SPAdes v3.15.3 [103] with default parameters. Gene models were predicted on repeat-masked assemblies using Funannotate v1.8.9 (https://github.com/nextgenusfs/funannotate), as previously described [30]. Manual inspection and curation were limited to the mating-type loci, including the annotation of short mating pheromone precursor genes. Functional annotation was performed using the Funannotate "*annotate*" module and incorporated PFAM and InterPro domains, Gene Ontology (GO) terms, COGs, CAZymes, fungal transcription factors, MEROPS proteases, secreted proteins, secondary metabolites, and BUSCO groups, as previously detailed [30]. All assemblies and corresponding raw sequencing data have been deposited in DDBJ/EMBL/GenBank under the BioProject numbers listed in S6 Table. The assemblies and annotations are also available at Figshare (https://doi.org/10.6084/m9.figshare.30375169).

**Ortholog identification, phylogenetic analyses, and topological support**

To construct the phylogenomic data matrix, single-copy orthologs (SC-OGs) were identified across all *Cryptococcus* species and strains, along with the outgroup *Kwoniella shandongensis* (GCA_008629635.2), using OrthoFinder v3.0.1b1, as previously outlined [30]. A total of 3,510 SC-OGs shared among all species were identified. To ensure quality and comparability, only proteins from the *C. neoformans* H99 reference strain that were ≥100 amino acids in length and whose orthologs exceeded 50% of the H99 protein length were retained. These filtering steps yielded 3,321 SC-OGs, which were individually aligned with MAFFT (L-INS-i mode) [104] and trimmed with TrimAl (gappyout) [105]. To enrich phylogenetic signal, only alignments with >30% parsimony-informative sites were retained using PhyKIT (*pis*) [106], resulting in a final dataset of 2,687 SC-OGs. Maximum likelihood (ML) and coalescence-based species trees were inferred from this dataset. For ML inference, SC-OG alignments were concatenated into a partitioned supermatrix (39 taxa, 2,687 partitions, and 1,625,459 sites) and analyzed with IQ-TREE using the "-*p*" option [107]. For coalescence-based inference, individual gene trees were inferred with IQ-TREE using the "-*S*" option. Branches with <70% UFBoot support were collapsed using Newick utilities (*nw_ed* "*i & b <= 70*" *o*), and a species tree was then estimated with ASTRAL v5.7.8. Branch support and genealogical concordance were assessed using SH-aLRT, UFBoot, gene concordance factors (gCF), and site concordance factors (sCF) for ML trees, and local posterior probabilities (LPP) and quartet support for the ASTRAL tree. Phylogenetic trees were visualized and annotated with iTOL v7. Detailed scripts containing software versions, and model parameters used in these analyses, including visualization of gCF/sCF and ASTRAL support, are accessible at https://doi.org/10.5281/zenodo.15686462. The complete set of ML gene trees that served as input for the different analyses, along with the quartet annotations file containing topology partitions and frequency data, is provided as S1 File.

**ANI, AAI, and dDDH**

Pairwise average nucleotide identity (ANI) values were computed using OrthoANIu [108]. Custom Python scripts were used to parse and reformat the resulting matrix, convert ANI values to distance scores (100 − ANI), and perform average linkage clustering to generate dendrograms, and Newick-format trees visualized with iTOL v7. Minimum, maximum, and mean ANI values between species or clade groupings were also calculated using a metadata file mapping strains to their

respective groups. Average amino acid identity (AAI) was computed using EzAAI v1.2.3 [109] with the DIAMOND algorithm and thresholds of 0.4 identity and 0.5 coverage. Pairwise AAI values were parsed using Python scripts and similarly used for clustering and for computing summary statistics across species or clades. Correlation analysis between pairwise ANI and AAI values was assessed using a custom Python script and by computing Pearson and Spearman correlation coefficients. Pairwise digital DNA–DNA hybridization (dDDH) values were estimated through the Genome-to-Genome Distance Calculator (GGDC 3.0; https://ggdc.dsmz.de/ggdc.php), using Formula 2, which is recommended for incomplete genome assemblies [39]. Due to the requirement for online submission, representative intra- and interspecies genome comparisons were selected and submitted manually (S2 Table). The output files were parsed with a custom Python script, which extracted Formula 2 dDDH values and the associated probability of exceeding the 70% species threshold. Based on these values, comparisons were categorized as intra- or interspecies, and further classified into confidence levels. This analysis was used to assess congruence with ANI and AAI-based classifications and evaluate species boundaries among closely related lineages. Detailed scripts containing software versions and parameters used in these analyses are available at https://doi.org/10.5281/zenodo.15686462.

## Synteny analyses and centromere identification

Synteny relationships were assessed using three complementary approaches. First, for full karyotype comparisons across selected species, we applied the ntSynt toolkit [110] to identify both strain- and species-specific chromosomal rearrangements (Figs 3A, 4H, S3A, and S4A). For clade A comparisons, ntSynt was run with the following parameters: *-d 5, -k 21, -w 200, --block_size 5000, --merge 20000, --indel 5000, --w_rounds 200 100*. For clade D comparisons, the following parameters were used: *-d 20, -k 24, -w 200, --block_size 1000, --merge 10000, --indel 5000, --w_rounds 200 100*. Resulting synteny blocks were visualized using ntSynt-viz [111], which combines ribbon-style plotting with chromosome painting. A pruned species tree in Newick format was supplied for each comparison to guide genome assembly order and improve visual coherence.

Second, to examine chromosomal structural conservation and intercentromeric translocations in greater detail, we performed nucleotide-level synteny analyses using a custom Python workflow (*0_synteny_karyoplotter.py*). Genome-wide alignments were generated with minimap2 (v2.17-r941; *-x asm20*), and synteny blocks were parsed from the resulting PAF files. Analyses were conducted in reference-based mode, with all assemblies aligned to a single genome: *C. neoformans* strain 125.91 for clade A comparisons (Fig 3B), and *C. porticicola* DSM108351 for clade D (Fig 5A). Centromere coordinates were provided in BED format and overlaid on the genome plots to highlight centromeric regions and infer intercentromeric rearrangements. All visualizations were generated in Python 3 using Biopython, pandas, matplotlib, and numpy, and the final SVG outputs were manually refined in Adobe Illustrator for publication.

Third, linear synteny plots comparing chromosomes and selected genomic regions in clade D species, particularly around centromeres, were generated with EasyFig v2.2.2 [112] using BLASTN for alignment. Centromere positions and genome-wide GC content were determined as previously described [30,32]. Centromeres were defined as the longest ORF-free regions enriched in centromere-associated LTR retrotransposons and exhibiting elevated 5mCG methylation, with boundaries set by conserved flanking centromeric genes. GC content and 5mCG methylation profiles were visualized in IGV and precisely overlaid at scale with EasyFig-derived synteny plots in Adobe Illustrator. Scripts are available at https://doi.org/10.5281/zenodo.15686462.

## DNA methylation detection

DNA methylation analysis was performed using a Dorado–Modkit workflow to detect 5-methylcytosine at CpG dinucleotides (5mCG) from Oxford Nanopore sequencing data. Raw signal data in POD5 format were basecalled with Dorado v0.5.2 using the combined model *sup@latest,5mCG_5hmCG@latest,6mA@latest* to simultaneously call canonical bases and modifications. For previously generated Nanopore datasets available only in FAST5 format, raw signals were first

converted to POD5 using the ONT `pod5 convert fast5` utility. Output BAM files were then realigned to the genome assemblies with the Dorado aligner and sorted with SAMtools v1.10. Methylation signals were extracted and quantified using Modkit v0.2.5-rc1 (pileup mode, `--preset traditional`) to generate both pileup and bedGraph outputs. Centromere-associated methylation profiles were visualized in IGV and plotted to scale in Adobe Illustrator for integration with genome structure and synteny plots.

**rDNA analysis**

A unified Python-based pipeline was developed for identifying and analyzing the distribution of rRNA genes across *Cryptococcus* genomes. For each genome, rRNA genes were annotated using Barrnap v0.9 (https://github.com/tseemann/barrnap) with the "`--kingdom euk`" setting, and coordinates for 5S, 5.8S, 18S, and 28S rRNA genes were extracted from the GFF output. Canonical rDNA clusters were defined as colocalized 28S–5.8S–18S genes on the same strand within a 5-kb window, optionally including an adjacent 5S gene. Per-genome chromosome plots were generated to display scaled chromosomes, centromeres, and rRNA gene positions. To test whether 5S genes were non-randomly distributed across chromosomes, the expected number of 5S genes per contig was estimated by normalizing contig lengths relative to the total genome size, assuming uniform probability of 5S placement. Observed versus expected values were compared using a chi-squared goodness-of-fit test, implemented with scipy.stats.chisquare. Standardized residuals were calculated to identify contigs with over- or under-represented 5S copy numbers, and results were tabulated per genome. Additionally, to assess 5S enrichment in subtelomeric regions, subtelomeric windows were defined as the first and last 2.5% of each contig, corresponding to 50 kb at each end of a 2 Mb chromosome. The midpoint of each 5S gene was used to determine overlap with these regions. A 2×2 contingency table was generated for each genome comparing the number of 5S genes located inside versus outside subtelomeric windows, relative to the corresponding base pair content. Significance was evaluated using a one-sided Fisher's exact test (scipy.stats.fisher_exact, alternative="greater") to test for enrichment. All gene coordinates, cluster counts, statistical results, and graphical outputs were exported as structured TSV and SVG files for each genome. To enhance figure clarity and readability, plots were refined by adjusting color schemes, modifying labels, and manually adding additional features using Adobe Illustrator. Scripts and raw data files are available at https://doi.org/10.5281/zenodo.15686462 and in S3 Table.

**Variant analysis and phylogenetic network in *C. porticicola***

Variant analysis within *C. porticicola*, was performed with Snippy v4.6.0 (https://github.com/tseemann/snippy), mapping reads to the strain DSM108351 reference genome (`--cpus 28, --unmapped, --mincov 10, --minfrac 0.9`). The resulting core SNP alignment (core.aln) served as input for Neighbor-net network reconstruction in SplitsTree v6.4.14. Recombination was assessed with the Pairwise Homoplasy Index (Phi) test, a permutation-based method implemented in SplitsTree that detects recombination by identifying statistically significant incompatibilities among adjacent sites in an alignment. The test compares observed homoplasy to that expected under a model of clonal evolution, with p-values < 0.05 interpreted as evidence for recombination [113].

**Population structure and recombination analyses in *C. porticicola***

To investigate recombination within *C. porticicola*, we developed a series of analysis scripts (with filenames indicated in parentheses and available at https://doi.org/10.5281/zenodo.15686462) beginning with the conversion of the multi-sample VCF generated by Snippy to PLINK binary format using PLINK v1.9 with the `--vcf` and `--allow-extra-chr` options (`0_convert_vcf_to_plink.sh`). Principal Component Analysis (PCA) was conducted in PLINK (`1_run_pca.sh`), and the resulting eigenvectors were plotted using Python (`2_plot_pca.py`) with pandas, seaborn, and matplotlib. To further examine signatures of historical recombination, linkage disequilibrium (LD) decay was assessed using PopLDdecay (`3_run_ld_decay_compare.sh`) with a maximum inter-SNP distance of 250 kb. To reduce the influence of rare alleles on

LD estimates, we applied a minor allele frequency (MAF) threshold of 0.167, corresponding to a 1/3 threshold for biallelic variants in haploid genomes (i.e., at least 2 of 6 strains carrying the minor allele). LD decay was also calculated without MAF filtering for comparison. The resulting files were parsed and plotted using Python (*4_plot_ld_decay.py*), applying a centered rolling mean and LOWESS smoothing to generate smoothed $r^2$ curves. LD50 values were estimated as the distance at which smoothed $r^2$ values dropped to 50% of the initial maximum, providing a quantitative summary of LD decay.

## Biolog phenotypic assays, scoring, cross-validation, and clustering analyses

Phenotypic MicroArray tests were performed using Biolog Gen III, FF, and YT microplates. Fresh 3–7-day-old cultures grown on PDA were used for plate inoculations. Cell density was adjusted to the manufacturer's recommended values for YT microplates, using sterile water for YT and FF plates and Inoculation Fluid B for Gen III plates. Microplates were inoculated with 100 µl of cell suspension per well and incubated in the dark at 25°C. Absorbance was measured immediately after inoculation and then every 3–7 days for up to 21 days at 490 nm, 590 nm, and 750 nm using a Varioskan LUX Multimode Microplate Reader (Thermo Fisher Scientific). Plates were shaken before each reading to ensure a homogeneous suspension.

Because Biolog assays are end-point measurements, wells showing color development or turbidity within the incubation period were scored as positive, whereas those showing no signal and values close to the negative control were scored as negative. Ambiguous cases were re-evaluated against standard assimilation profiles for *Cryptococcus* species and, when possible, retested in traditional liquid or microplate formats following the methods described in Kurtzman et al. (2011) [114]. Because growth dynamics differ between simple and complex substrates, a key challenge in assessing growth was establishing a reliable baseline to distinguish positive from negative growth responses. To ensure robust interpretation across substrates of differing complexity, we implemented a cross-validation framework drawing on published physiological data [114] and culture-collection databases (CBS WI, PYCC), and by repeating a subset of tests in tubes or custom microplates. Cross-validation covered 12 of the 17 recognized species (71%), including 50% of clade A (*C. neoformans*, *C. deneoformans*, *C. bacillisporus*, *C. gattii*, and *C. hyracis*) and clade B (*C. cederbergensis*) species, and all species from clades C (*C. floricola*, *C. wingfieldii*, *C. amylolentus*, and *C. maquisensis*) and D (*C. porticicola*, *C. ficicola*, and *C jaspeensis*). In total, reference data were available for 46 of the 153 substrates tested (30% overall, corresponding to 49–77% of a single plate) and included representatives of monosaccharides, disaccharides, polysaccharides, polyols, organic acids, and low-molecular-weight aromatic compounds (S4C Table). When possible, members of the same clade were processed in parallel, which helped to flag and resolve occasional inconsistencies. These comparisons informed interpretation thresholds and confirmed that the majority of Biolog calls were reproducible across methods.

Growth on glycerol and xylitol was further assessed by spot assays on solid Yeast Nitrogen Base (YNB) medium without amino acids. A 10 × YNB stock was prepared by dissolving 6.7 g of YNB without amino acids and the appropriate carbon source in 100 ml of water, adjusting the pH to 5.6, and sterilizing by filtration. Each medium was normalized to provide the same total carbon as 5 g of glucose [114]. For each 100 ml of final medium, 10 ml of 10 × YNB + carbon source was mixed with 90 ml of a 2% agar solution (autoclaved and cooled to ~50°C). Glycerol and xylitol were tested alongside glucose (positive control). YNB without added carbon served as a control for potential nutrient carry-over (negative control). Inocula were prepared by growing the cells for 48 h at room temperature (25 ± 1°C) in 5 ml of YNB + 0.1% glucose (carbon-starvation medium) at 70 rpm [114]. Cells were pelleted, washed twice, adjusted to an optical density of 2 at 600 nm, serially diluted 10-fold, and 3 µl of each dilution was spotted onto the plates. Growth was assessed after 7 days at room temperature (25 ± 1°C).

Phenotypic data from Biolog assays were analyzed with a custom Python pipeline (*0_biolog_analysis_pipeline.py*). Input consisted of a manually curated Excel file (S4 Table) containing binary growth data across multiple *Cryptococcus* strains. Dimensionality reduction and clustering analyses were performed to explore global patterns of phenotypic similarity. Principal component analysis (PCA) was conducted on the standardized dataset, and the first two components were

visualized with 95% confidence ellipses grouped by clade. A binary clustermap was generated using hierarchical clustering based on Euclidean distance and average linkage. To identify traits most informative for clade-level classification, a Random Forest classifier was trained on the binary matrix treating clade membership as the target variable, and the top 15 features ranked by importance were retained. Clade-wise presence frequencies for these phenotypes were then calculated and visualized. Redundant phenotypes, defined as those with identical binary profiles across all strains, were automatically flagged to reduce overrepresentation. All analyses and visualizations were performed using Python with pandas, scikit-learn, seaborn, matplotlib, and related packages.

## Standard physiological tests

For standard species descriptions, growth tests were also performed in liquid media using both tubes and microplates. Tube tests were conducted in 2 ml volumes in 5- and 7.5-ml plastic tubes following the method described by Kurtzman et al. (2011) [114]. Microplate tests were performed in 200 µl volumes using the same media composition as the tube tests. Tubes and plates were shaken every 1–2 days. Growth in tubes was evaluated visually, while growth in microplates was measured at 600 nm using the Varioskan LUX microplate reader. Results were scored as positive or weak when there was clear or reduced growth compared to glucose, and as negative when growth was minimal or absent relative to both the negative control and glucose. The custom microplate format also enabled cross-validating of some substrates (e.g., L-sorbose, D-glucosamine, and D-saccharic acid) or testing of additional substrates (e.g., dulcitol, soluble starch, and aldaric acids such as D-tartaric, and L-tartaric), as well as growth under osmotic stress (NaCl 5%, 8%, and 10%) and in high-glucose conditions (50% and 60%).

## Temperature and melanin production assays

For temperature assays, pre-inocula were grown in 4 ml of YPD medium overnight at room temperature. New tubes containing 4 ml of YPD were inoculated with 50 µl of each overnight-grown pre-inocula and incubated at room temperature for 48 hours. Cells were then pelleted, washed with sterile distilled water, and adjusted to an optical density of 2.0 at 600 nm. Serial 10-fold dilutions were prepared from these normalized cultures, and 3 µl of each dilution was spotted onto YPD agar plates. Plates were incubated at 23°C, 30°C, 35°C, and 37°C, and photographed after 3 days. For melanin production assays, 3 µl of each dilution was also spotted onto two melanin-inducing media: L-DOPA agar (7.6 mM L-asparagine, 5.6 mM glucose, 22 mM $KH_2PO_4$, 1 mM of $MgSO_4.7H_2O$, 0.5 mM L-DOPA, 0.3 µM thiamine, 20 nM biotin and 20 g/L of agar) and Bird seed agar (7% bird seed extract, 0.1% glucose, and 20 g/L agar) [101,115]. Bird seed agar provides a complex, plant-derived substrate rich in natural phenolic precursors (e.g., caffeic acid and related catechols) that are oxidized by fungal phenoloxidase (laccase), producing the characteristic brown-black pigmentation. In contrast, L-DOPA medium supplies a single defined precursor, allowing a direct test of laccase-dependent melanin synthesis under controlled conditions. A *C. neoformans* H99 *lac1Δ* mutant was used as a negative control. Plates were incubated at room temperature in the dark for 5 days before imaging. All assays were performed in three independent biological replicates to ensure reproducibility of phenotypic results.

### Environmental detection via GlobalFungi database

To screen for environmental presence of the newly described *Cryptococcus* species, we performed exact-match searches against the GlobalFungi release 5 database [57], which compiles ITS1 and ITS2 amplicon data from 84,972 samples across 846 studies. Full-length ITS1 and ITS2 regions were extracted from each genome using ITSx [116] implemented in SEED 2 [117]. Unique haplotypes were identified through dereplication in SEED 2, and only sequences identical in both length and nucleotide composition were considered matches. ITS sequences from additional *Cryptococcus* species included in the phylogenetic analysis (Fig 1) were queried for comparison. Full metadata and hit summaries are available in S5 Table.

## Supporting information

**S1 Fig. Expanded view of phylogenomic support and conflict among clade A *Cryptococcus* species. (A)** Pruned maximum likelihood (ML) phylogeny (from Fig 1A), showing only clade A species. The tree was inferred from a concatenated alignment of 2,687 single-copy orthologs (SC-OGs). Conflict-prone nodes are marked in red and labeled by node ID. Concordance factors based on gene trees (gCF) and site patterns (sCF) are shown in grey. All nodes are fully supported by SH-aLRT and ultrafast bootstrap (UFboot = 100%). **(B)** Pruned ASTRAL species tree (from Fig 1B), based on the same SC-OGs. All nodes have local posterior probability (LPP) = 1.0. Quartet support for the main topology is shown in grey. Nodes corresponding to those in panel A are labeled by their ASTRAL node IDs. **(C)** Gene and site concordance at three conflict-prone nodes (N47, N52, N61). Gene concordance factors (gCF) and discordance factors (gDF1, gDF2, and gDFP) are shown on the left. The main topology is shown in dark blue; alternatives and paraphyly are shown in lighter blue shades. Site concordance (sCF) and discordance (sDF1, sDF2) are shown on the right. The red dashed line at 33.3% represents the expected support under a hard polytomy. **(D)** ASTRAL quartet support for the corresponding nodes (N28, N23, N16). Bar plots show the normalized frequency of each of the three possible quartet topologies, with the main topology shown in dark green and the two alternatives in lighter shades. Local posterior probabilities (LPP) are indicated above each bar. In all cases, ASTRAL strongly supports the main topology (LPP = 1.0), despite the presence of gene tree discordance in the ML-based analyses (panel C). This highlights how coalescent-based methods can recover consistent species relationships even in the presence of substantial incomplete lineage sorting, which likely underlies the low gCF and high gDFP values observed at these nodes.
(TIF)

**S2 Fig. Mating assays demonstrate the ability of *Cryptococcus hyracis* to undergo sexual reproduction. (A)** Pairwise mating assays using two *C. hyracis MAT*α strains (MF34 and MF13) crossed with *MAT***a** strains from other species in the *C. gattii* complex. Crosses were performed on V8 agar and examined by light microscopy after 2–3 weeks. Hyphal growth with basidia and spores (inset) indicated successful mating; positive and negative interactions are marked with "+" and "–", respectively. Strain MF13 generally showed stronger mating responses than MF34. For *C. bacillisporus*, we tested both the wild-type strain B4546 and a derived *crg1*Δ mutant. *CRG1* encodes a negative regulator of pheromone signaling that acts via G-protein signaling by stimulating GTP hydrolysis to GDP, thereby extinguishing Gα-GTP signaling. Thus, deletion mutants are hypersensitive to mating signals and serve as sensitive indicators of mating potential. Scale bars = 200 µm (10 µm in insets). **(B)** Positive control cross between *C. neoformans MAT*α (KN99α) and *MAT***a** (KN99**a**) showing robust filamentation and basidia formation. **(C–E)** Scanning electron microscopy (SEM) of interspecies mating interactions involving *C. hyracis*. Panels show examples of basidia and basidiospore formation, as well as fused clamp connections, observed in crosses with *C. bacillisporus* (C) and *C. tetragattii* (D–E). These structures confirm the ability of *C. hyracis* to engage in sexual development.
(TIF)

**S3 Fig. Karyotype comparisons among *C. neoformans* and *C. deneoformans* strains. (A)** Synteny analysis of 14 chromosomes across multiple *C. neoformans* strains representing the four major lineages (VNI, VNII, VNBI, VNBII) and two *C. deneoformans* (VNIV) strains. Chromosomes were ordered to maximize collinearity with the *C. neoformans* strain 125.91 reference genome, and those inverted relative to their original orientations are indicated with asterisks. Overall synteny is well conserved across *C. neoformans* strains, except for strain H99 (VNI), which harbors a known translocation between chrs. 3 and 11. *C. deneoformans* strains are largely syntenic to *C. neoformans*, except for some large inversions. The locations of the rDNA cluster and *MAT* locus are indicated. Centromere positions are marked by grey triangles. Rearrangements, including large inversions and translocations, were identified using the ntSynt toolkit and visualized with ntSynt-viz. **(B)** Schematic of the previously characterized reciprocal translocation in *C. neoformans* H99 between chrs. 3 and 11. Dashed lines indicate the breakpoint positions, and circles mark centromere locations. This rearrangement distinguishes H99 from other VNI strains.
(TIF)

**S4 Fig. Karyotype comparisons among *C. gattii* (sensu lato) species. (A)** Synteny comparisons between *C. gattii* strain WM276 (VGI, reference) and representative strains from each of the species within the *C. gattii* complex. Chromosomes were reordered to maximize collinearity with the reference, and those inverted relative to their original assembly orientations are marked with asterisks. Synteny blocks were identified using the ntSynt toolkit and visualized with ntSynt-viz. Despite broad conservation of synteny, numerous rearrangements are observed across species, including translocations and inversions. The chromosomal locations of the rDNA cluster and the *MAT* locus are also indicated, as are predicted centromeres (triangles). **(B)** Schematic representations of selected interspecific chromosomal rearrangements. Colored bars represent putative ancestral chromosomes and their rearranged derivatives in extant species, with breakpoints indicated by dashed lines. Centromeres are shown as circles. Panels highlight key rearrangements across different nodes of the phylogeny, including several likely intercentromeric recombination (ICR) events, such as that shared between *C. bacillisporus* and *C. decagattii* (event b) and the derived translocation found in *C. tetragattii* CBS11718 (event f). (TIF)

**S5 Fig. Mating assays between *C. porticicola* strains under two different media conditions. (A)** Summary table indicating presence (+?) and absence (–) of hyphal and basidial-like structures. "+?" denotes putative structures consistent with sexual development, though the identity of basidia-like structures could not be definitively confirmed. Mating-type alleles (*STE3* and *HD*) are shown for each strain. **(B)** Representative light microscopy images from V8 pH 5.0 medium assays. Only the DSM108352 (a1b3) × NCYC1536 (a2b2) cross showed hyphal growth and potential sexual structures embedded in the agar (arrow); the inset highlights a structure resembling a basidium. Scale bars: 200 μm (50 μm for inset). **(C)** Summary table of mating outcomes on MS medium. **(D)** Representative light microscopy images from MS medium assays. Pseudohyphal growth, characterized by constricted septa and chain-like elongation, is observed in the solo culture of NCYC1536 and in the DSM108351 × NCYC1536 cross (arrows). True hyphae growth, defined by continuous non-constricted filaments, is observed in the DSM108352 × NCYC1536 cross (arrow), although no apparent basidial-like structures were detected throughout the course of the experiment (marked as "–*"). Scale bars: 200 μm unless otherwise noted (50 μm for insets). (TIF)

**S6 Fig. Synteny and structural context of conserved and apparently lost centromeres across clade D *Cryptococcus* species. (A–E)** Synteny comparisons across conserved centromeric regions in *C. porticicola* (strains DSM111204, DSM108352, NCYC1536, and DSM108351), *C. ficicola* (DSM109063), and *C. jaspeensis* (OR918). Tracks highlight conserved centromere positioning and surrounding synteny for *CEN1*, *CEN2*, *CEN3*, *CEN4*, and *CEN6* of *C. porticicola* DSM108351 and their corresponding orthologous regions in the other strains and species. **(F)** Apparent loss of *CEN7* in *C. jaspeensis* OR918. The top panel shows chromosome-scale synteny across strains, highlighting the conserved position of *CEN7* in all species except *C. jaspeensis*, where the orthologous region lacks canonical centromeric signatures and is marked as putative inactivated centromere (*iCEN*). The bottom panel provides a zoomed-in view of this region with tracks showing synteny, 5mCG DNA methylation, and TE content. Although the centromeric repeat landscape appears disrupted in OR918, residual DNA methylation and partial synteny may indicate a recently inactivated centromere. **(G)** Apparent centromere loss in both *C. ficicola* and *C. jaspeensis*. The top panel shows chromosome-scale synteny, highlighting the position of *CEN8* from *C. porticicola* DSM108351. In strain DSM111204, this region underwent intercentromeric recombination, while in *C. ficicola* and *C. jaspeensis* the corresponding region lacks the repeat-rich structure typical of active centromeres. The bottom panel shows a zoomed-in view of this region, indicating that the orthologous loci in DSM109063 and OR918 lack the repeat-rich structure characteristic of active or ancestral centromeres, and only residual 5mCG DNA methylation, consistent with inactivated centromeres (*iCEN*). (TIF)

**S7 Fig.  Genome-wide organization of ribosomal RNA (rRNA) genes across *Cryptococcus* species.** Chromosomal distribution of 28S, 5.8S, 18S, and 5S rRNA genes across all analyzed *Cryptococcus* strains with chromosome-level assemblies. Arrowheads indicate the position of the core rDNA array, and predicted centromeres are marked with open circles. In *C. floricola*, chr. 7 is represented as two joined contigs with a gap at *CEN7* (marked with an asterisk). The rDNA array (28S–5.8S–18S) is typically clustered on a single chromosome, whereas 5S genes show variable distribution patterns across lineages. **(A)** Clade A (pathogenic) species exhibit fully clustered rRNA arrays, while saprobic species from other clades **(B-D)** display unclustered configurations with lineage-specific features: scattered 5S genes (e.g., *C. depauperatus* and *C. porticicola*), chromosome-specific enrichment (e.g., *C. floricola*), or subtelomeric localization (e.g., *C. jaspeensis*). In *C. amylolentus*, the 28S–5.8S–18S rDNA cluster is absent from the current assembly owing to unresolved repetitive regions near the end of chr. 9. The inferred position, marked with an arrowhead labeled "(a)," is supported by a few long reads from the original sequencing dataset that extend into the rDNA region.
(TIF)

**S8 Fig.  Phenotypic traits distinguishing *Cryptococcus* species and clades. (A)** Top 15 phenotypic tests contributing most to clade-level classification based on Random Forest analysis of binarized Biolog data. Substrates are ranked by feature importance in predicting clade membership. **(B)** Mean presence frequency of the top 15 substrates from panel A across *Cryptococcus* clades. Some substrates (e.g., glycerol, xylitol, glucuronamide) display clade-specific patterns, with utilization largely restricted to or absent from particular clades, highlighting their potential utility for phenotypic discrimination. **(C)** Growth of all tested strains at 23°C, 30°C, 35°C, and 37°C on YPD medium for 4 days. Cells were grown for 48 hours in YPD, washed, and spotted in 10-fold dilutions starting from an OD600 of 2.0. Robust growth at 37°C was observed exclusively in clade A strains, consistent with thermotolerance as a pathogenicity-associated trait. No strains from saprobic species in clades B–D showed appreciable growth at this temperature. Among saprobes, *C. cederbergensis* displayed limited growth above 30°C, whereas *C. maquisensis* OR849 exhibited weak growth at 35°C (visible in the highest inoculum spot). A colony marked with an "X" represents contamination and should be disregarded. **(D)** Melanin production on L-DOPA agar plates incubated at room temperature for 5 days. Spotting assay was conducted as described in C. All clade A strains produced visible melanin, while all nonpathogenic strains from clades B–D lacked pigmentation under these conditions. Among pathogens, melanin production was somewhat reduced in *C. decagattii* CBS11687 and *C. deneoformans* NIH12 under this growth condition.
(TIF)

**S9 Fig.  Validation of glycerol and xylitol assimilation across *Cryptococcus* species.** Spot assays testing growth on defined minimal medium (YNB without amino acids) containing glucose, glycerol, or xylitol as the sole carbon source. YNB without added carbon served as a control for potential nutrient carry-over. Cells were pre-grown for 48 h at room temperature (25 ± 1°C) in YNB + 0.1% glucose (carbon-starvation medium), washed, and spotted in 10-fold dilutions starting from an OD600 of 2. Plates were incubated at room temperature for 7 days before imaging. Pathogenic clade A species exhibited little or no growth on glycerol or xylitol, whereas non-pathogenic species (clades B–D) grew well under the same conditions.
(TIF)

**S1 Table.  List of strains used in this study, and summary of genome assembly statistics and other genomic features.**
(XLSX)

**S2 Table.  ANI, AAI, and dDDH metrics across *Cryptococcus* genomes. (A)** Full matrix of ANI values computed for all pairwise genome comparisons. **(B)** Summary of pairwise ANI values between species. **(C)** Min., max., and mean ANI values calculated between and within clades, based on grouped pairwise comparisons. **(D)** Full AAI values computed for

all pairwise genome comparisons. **(E)** Summary of pairwise AAI values between species. **(F)** Min., max., and Mean AAI values calculated between and within clades, based on grouped pairwise comparisons. **(G)** Closest interspecies genome comparisons based on identity values. **(H)** Digital DNA-DNA hybridization (dDDH) values from Genome-to-Genome Distance Calculator (GGDC). **(I)** dDDH values with species-level summary and strain annotations.
(XLSX)

**S3 Table. rRNA gene analysis and statistical tests. (A)** Genome-wide summary of rRNA gene counts. **(B)** Genome-wide summary of 5S rRNA gene counts. **(C)** Chromosomal distribution of 5S rRNA genes, observed vs. expected. **(D)** Chromosomal enrichment of 5S rRNA genes and statistical analyses. **(E)** Summary of 5S gene localization relative to subtelomeric regions.
(XLSX)

**S4 Table. Phenotypic profiling results. (A)** Full list of tests. **(B)** Informative tests showing variable response across strains. **(C)** Cross-validation of assays.
(XLSX)

**S5 Table. Summary of GlobalFungi ITS matches to newly described *Cryptococcus* species.** Table listing exact ITS1 and ITS2 matches identified in the GlobalFungi database for each *Cryptococcus* species. Includes marker resolution (delimitation power) of ITS1 and ITS2 across species, match type (ITS1 or ITS2), geographic origin, substrate, and study metadata. Data correspond to matches visualized in S2 File.
(XLSX)

**S6 Table. Genome sequencing and information on raw sequencing data generated in this study. (A)** Genome sequencing, assembly, polishing, and gene prediction/annotation approaches. **(B)** NCBI accession numbers of each genome and raw read data generated and used in this study.
(XLSX)

**S1 File. Gene tree and quartet topology datasets used for concordance-factor and ASTRAL analyses.** Includes ML gene trees (trimal_pergene.treefile), trees with branches <70% UFboot collapsed for ASTRAL (trimal_pergene.BS70.treefile), and the ASTRAL quartet annotation file with topology partitions and frequencies (trimal_pergene.t16.freqquad.csv).
(ZIP)

**S2 File. Interactive map of GlobalFungi sampling sites.** Interactive HTML map showing GlobalFungi sampling sites with exact ITS1 or ITS2 matches to newly described *Cryptococcus* species, as summarized in S5 Table. The interactive distribution map was produced using the leaflet package in R v2.0.4.1, and the base layer (country and region borders) was sourced from Natural Earth (http://www.naturalearthdata.com/), which provides public domain map data compatible with the CC BY 4.0 license (http://www.naturalearthdata.com/about/terms-of-use/).
(ZIP)

## Acknowledgments

We thank Dr. Seonju Marincowitz and Dr. Nam Pham for their assistance in isolating *Cryptococcus cederbergensis*, Dr. Tatiana A. Kuznetsova for assistance in isolating *C. porticicola* strains, and Dr. Oliver Röhl for assistance in isolating strains OR849 (*C. maquisensis*) and OR918 (*C. jaspeensis*). We are indebted to Rytas Vilgalys for inspiring the study of non-pathogenic *Cryptococcus* species, and to Brenda Wingfield for forging connections that greatly contributed to making these advances possible. We also thank Dr. K.J. Kwon-Chung for her foundational contributions to the study of *Cryptococcus* species and for laying the groundwork for the description of the VGV lineage now formally named in this study. We

are grateful to Dr. Konstanze Bensch for her valuable suggestions and help with the Latin naming of the new species. We also thank Dr. Fred Dietrich for providing computational resources.

## Author contributions

**Conceptualization:** Marco A. Coelho, Márcia David-Palma, Joseph Heitman.

**Data curation:** Marco A. Coelho, Márcia David-Palma, Andrey M. Yurkov.

**Formal analysis:** Marco A. Coelho, Miroslav Kolařík.

**Funding acquisition:** Michael J. Wingfield, Andrey M. Yurkov, Joseph Heitman.

**Investigation:** Marco A. Coelho, Márcia David-Palma, Andrey M. Yurkov.

**Project administration:** Joseph Heitman.

**Resources:** Aleksey V. Kachalkin, Miroslav Kolařík, Benedetta Turchetti, José Paulo Sampaio, Michael J. Wingfield, Matthew C. Fisher, Andrey M. Yurkov, Joseph Heitman.

**Software:** Marco A. Coelho.

**Supervision:** Joseph Heitman.

**Visualization:** Marco A. Coelho, Márcia David-Palma.

**Writing – original draft:** Marco A. Coelho, Márcia David-Palma.

**Writing – review & editing:** Marco A. Coelho, Márcia David-Palma, Aleksey V. Kachalkin, Miroslav Kolařík, Benedetta Turchetti, José Paulo Sampaio, Michael J. Wingfield, Matthew C. Fisher, Andrey M. Yurkov, Joseph Heitman.

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
