## [Decision Letter · Decision Letter 0]

22 Sep 2025

PGENETICS-D-25-00869

Genomic and phenotypic insights into the expanding phylogenetic landscape of the Cryptococcus genus

PLOS Genetics

Dear Marco / Dr. Coelho,

Thank you for submitting your manuscript to PLOS Genetics.

Two experts reviewed your manuscript have identified a list of straightforward modifications that you should address prior to acceptance of your manuscript. I ask that you pay particular attention to addressing Reviewer #1's request for "a clearer synthesis of the data as a whole in the discussion, specifically with regard to how to go about defining what a species is and how the authors are integrating the datasets they collected here (or not)" and Reviewer #2's request to "tone down ...[your] language with respect to "centromere inactivation"".

Please submit your revised manuscript within 30 days Oct 22 2025 11:59PM. If you will need more time than this to complete your revisions, please reply to this message or contact the journal office at plosgenetics@plos.org. Please include the following items when submitting your revised manuscript:

We look forward to receiving your revised manuscript.

Kind regards,

Antonis Rokas

Guest Editor

PLOS Genetics

Geraldine Butler

Section Editor

PLOS Genetics

Aimée Dudley

Editor-in-Chief

PLOS Genetics

Anne Goriely

Editor-in-Chief

PLOS Genetics

**Additional Editor Comments:**

Reviewer #1:

Reviewer #2:

**Journal Requirements:**

4) Please send a completed 'Competing Interests' statement, including any COIs declared by your co-authors. If you have no competing interests to declare, please state "The authors have declared that no competing interests exist". Otherwise please declare all competing interests beginning with the statement "I have read the journal's policy and the authors of this manuscript have the following competing interests"

**Reviewers' comments:**

Reviewer's Responses to Questions

**Comments to the Authors:**

Reviewer #1: This study reports new genomic characterization of 6 species within the Cryptococcus lineage and places them within a revised species phylogeny for this genus. The authors perform extensive genomic analysis of 17 total Cryptococcus species looking not just at sequence diversity but also at chromosomal rearrangements, karyotype and centromere evolution and rRNA array structure. In addition, they provide phenotypic characterization of mating between strains and species and growth of these species on different substrates. In all, I found this to be a compelling and comprehensive story investigating an important genus of fungi and highlighting interesting themes in genome evolution and the origins of pathogenicity. I think this will be of exceptional interest to those in the Cryptococcus field, but the broad topics covered will likely make this also appealing to a wider audience. That said, I do have a few concerns, questions, and clarifications outlined below:

Major Comments

-The authors discuss species concepts/speciation throughout, but I came away from this paper confused about how they would actually like me to interpret what a “species” is and how to define species boundaries. They start to lay out how it can be powerful to take a holistic approach incorporating many different types of data in these decisions including: nucleotide/sequence diversity, chromosomal diversity, rDNA structure, mating, and metabolism. But then they launch into descriptions of the data and I am missing how they propose to incorporate all of this data into the decision of where to draw species boundaries? I don’t disagree with the distinctions they draw in the paper, but among their test cases are: C. hyracis vs. C. tetragattii, which they call separate species while having high(er) sequence divergence, low chromosomal structure/rDNA divergence, similar metabolism, and seem to mate (though knowing productivity seems very important here!) and the C. porticicola strains, which they group together as a single species, that have low sequence divergence, low karyotypic diversity, similar metabolism and largely don’t mate with each other. These only really seem to differ in sequence divergence and mating capacity. Does this suggest that the authors think sequence level diversity is the most useful in defining species boundaries?

Again, I have no real issue with the phylogeny they propose and I liked how they presented data on the many different axes of genomic/biological diversity that might contribute to speciation and reproductive isolation. But there’s both inter- and intra-species variation across all of these axes. I think my confusion here would largely be addressed just by a clearer synthesis of the data as a whole in the discussion, specifically with regard to how to go about defining what a species is and how the authors are integrating the datasets they collected here (or not). I do also think analysis of the viability of progeny in the matings they show would add significantly to this manuscript and to their discussion of speciation.

- It seems premature to make assumptions about the metabolic profiles of the species based solely on the biolog data without validating through other methods. And in general, the phenotyping section of this paper seemed a bit preliminary compared to the rest of the manuscript. It would be nice if the authors could validate growth/no growth phenotypes (or perform more nuanced growth assessments) by another method for at least the substrates pulled out by the random forest model. Maybe they already did this, but I found the methods and discussion of the auxanographic tube tests to be challenging to parse and this was not mentioned anywhere in the main text.

I was also particularly interested in the glycerol result. Namely, that clade A does not grow on glycerol. There are several published papers (PMCIDs: PMC2730461, PMC3110414, PMC8092316) which all show fairly robust growth for the C. neoformans reference strain (H99) on glycerol. I’m curious how the authors interpret this discrepancy?

Lastly, on this point, I’m curious if there are any logical explanations for how the most informative growth features correlate with distinct environments. For instance, are natural sources of xylitol correlated with the isolation sites of species that grow on it? Again it might be useful to have another complementary experiment for some of these metabolic traits that both confirms the biolog data but also potentially gives a more nuanced view of differences in growth. I don’t think this is essential, but it might help to strengthen the argument that ecological adaptation underlies some of the discordance between the metabolism/phenotypic data and the phylogenetics.

Minor Comments/Questions:

-lines 156-157 and throughout: How were C.neoformans strains chosen? Do they incorporate the maximum diversity available in strains of this species? This could be useful information to provide. Also seems important to consider the relative relatedness given the authors use of these strains to determine maximum intraspecies divergence--e.g. how much would the AAI and ANI and dDDH data change with different C. neoformans strain selections? I doubt it would change any interpretations but could be useful to discuss or compare.

-lines 413-414: It was very unclear to me until I got to the methods that the methylation analysis used in defining centromere locations was performed by the authors? Was all of the methylation data presented from this study? It would be useful to clarify what data was collected in this study vs. what came from published/other datasets (if anything).

-lines 915-917: I find the conservation of melanin biosynthesis genes across species very intriguing. Are there also transporters that are required for taking up melanin pre-cursors (e.g. L-DOPA)? Are these melanization genes/laccases used for other purposes? Are all of these predicted genes functional (i.e. have any been pseudogenized)?

-Figure S7C: It appears from the figure that C. amylolentus has no core rDNA array and no 28S-5.8S-18S sequences anywhere in the genome? This doesn’t seem correct?

Reviewer #2: The manuscript "Genomic and phenotypic insights into the expanding phylogenetic landscape of the Cryptococcus genus" by Coelho et al. presents an in depth examination of genomic and phenotypic diversity within the important fungal genus Crpytococcus. This includes the formal description of novel species as well as numerous intriguing observations about various facets of genome evolution in this group, much of which has important implications for the virulence of many of the strains. The manuscript is in excellent shape and can be published as is in my opinion. The authors have done a fantastic job with all of the analyses, presentations of results, and discussions. I'm particularly pleased to see the ANI analysis applied to species concepts here as this is something that is sorely needed in fungal taxonomy/systematics. My only overall critique is that the authors should tone down their language with respect to "centromere inactivation". Without data showing the location of the centromeric histones, the results, while compelling, are inconclusive. In particular I feel that lines 49/50 in the Abstract is too assertive given the data. Beyond this I only have minor (mostly pedantic) points that the authors could consider for a revised version, if they choose.

I like the historical context in the intro, maybe add to the opening paragraph that many species formerly called Cryptococcus have been delimitated to new genera.

Figure 1A has C. hyracis as a pathogen, but this is hypothetical, correct?

Is Figure 1C needed in a main figure? This type of workflow is more or less standard now for phylogenomic analyses, unless I am missing some novel approach?

For Figure 1D & E are the alternative topologies visualized somewhere? Same Question for Figure S1.

Maybe mention that C. neoformans and C. deneoformans can hybridize despite considerable evolutionary distance.

Line 303 should site previous literature that identified rearrangements among the Cryptococcal species.

Figure S5D the "pseudohyphae" and hyphae all look the same to me, and I would consider the pseudohyphae to look like true hyphae. Is there more rational for the differentiation beyond just degree of filamentation? In my experience the solo culture of DSM111204 is more typical of pseudohyphae.

Figure 4C I would say those structures are much more characteristic of chlamydospores than basidia, particularly with the thickness of the cell wall, but difficult to tell from these images.

Figure 4E should have percent variation on axes.

Figure 4G and corresponding results/discussion: Personally, I'm not convinced the four gamete test is appropriate for genome scale data. I would much prefer to compare to the LD50 values calculated in "The frequency of sex in fungi" by Nieuwenhuis & James. 13.9 kb would be halfway between S. cerevisiae and S. pombe and despite the authors claims in the discussion, is considerably higher than C. neoformans. As S. cerevisiae was estimated to undergo outcrossing at 1 in 10^5 generations and as S. pombe is thought to only undergo selfing (complicated by ancient admixture), this would suggest that mating (or at least outcrossing) in C. porticicola is exceptionally rare or only historic.

Some background on what centromeres look like in Cryptococcus could be helpful, as some readers may expect them to be point centromeres like in S. cerevisiae.

The putative neocentromere is extremely interesting, but speculative at this point. This could just be a new TE island. It would be great to see follow up work on this given how closely related the strains with different karyotypes are.

The rapid change in 5S architecture is also very intriguing. In Schizosaccharomyces, 5S evolution is linked to meiotic drive and the selfish wtf genes. This could be interesting to include in the discussion, although no meiotic drive has ever been found in a basidiomycete to my knowledge.

The section "Environmental detection of newly described species and limits of ITS-based detection" seems a bit tacked on. Some justification for why it is included would be nice or otherwise better integration to the rest of the manuscript.

**Have all data underlying the figures and results presented in the manuscript been provided?**

Reviewer #1: **No: ** Seems like currently the sequencing datasets are still processing on NCBI. The BioProjects listed exist but there are no datasets in them.

Reviewer #2: Yes

PLOS authors have the option to publish the peer review history of their article (what does this mean?). If published, this will include your full peer review and any attached files.

Reviewer #1: No

Reviewer #2: **Yes: ** Aaron A. Vogan

**Figure resubmission:**
---

## [Editor Report · Decision Letter 1]

30 Oct 2025

Dear Dr Coelho,

We are pleased to inform you that your manuscript entitled "Genomic and phenotypic insights into the expanding phylogenetic landscape of the Cryptococcus genus" has been editorially accepted for publication in PLOS Genetics. Congratulations!

Yours sincerely,

Geraldine Butler

Section Editor

PLOS Genetics

Aimée Dudley

Editor-in-Chief

PLOS Genetics

Anne Goriely

Editor-in-Chief

PLOS Genetics

BlueSky: @plos.bsky.social

Comments from the reviewers (if applicable):

**Data Deposition**

http://datadryad.org/submit?journalID=pgenetics&manu=PGENETICS-D-25-00869R1

**Press Queries**

---

## [Editor Report · Acceptance letter]

PGENETICS-D-25-00869R1

Genomic and phenotypic insights into the expanding phylogenetic landscape of the Cryptococcus genus

Dear Dr Coelho,

We are pleased to inform you that your manuscript entitled "Genomic and phenotypic insights into the expanding phylogenetic landscape of the Cryptococcus genus" has been formally accepted for publication in PLOS Genetics! Your manuscript is now with our production department and you will be notified of the publication date in due course.

With kind regards,

Zsofia Freund

PLOS Genetics

On behalf of:
